# BALANCING ACT: CONSTRAINING DISPARATE IMPACT IN SPARSE MODELS

**Meraj Hashemizadeh**[*1]  **Juan Ramirez** [*12]  **Rohan Sukumaran**[12]
**Golnoosh Farnadi**[134]  **Simon Lacoste-Julien**[124]  **Jose Gallego-Posada**[12]

[1]Mila  [2]DIRO - Université de Montréal  [3]McGill University  [4]Canada CIFAR AI Chair

## ABSTRACT

Model pruning is a popular approach to enable the deployment of large deep learning models on edge devices with restricted computational or storage capacities. Although sparse models achieve performance comparable to that of their dense counterparts at the level of the entire dataset, they exhibit high accuracy drops for some data sub-groups. Existing methods to mitigate this disparate impact induced by pruning (i) rely on surrogate metrics that address the problem indirectly and have limited interpretability; or (ii) scale poorly with the number of protected sub-groups in terms of computational cost. We propose a constrained optimization approach that *directly addresses the disparate impact of pruning*: our formulation bounds the accuracy change between the dense and sparse models, for each sub-group. This choice of constraints provides an interpretable success criterion to determine if a pruned model achieves acceptable disparity levels. Experimental results demonstrate that our technique scales reliably to problems involving large models and hundreds of protected sub-groups.

## 1 INTRODUCTION

Current deep learning practice displays a trend towards larger architectures (Bommasani et al., 2021), as exemplified by popular models such as GPT-4 (OpenAI, 2023), Llama 2 (Touvron et al., 2023) and DALL-E 2 (Ramesh et al., 2022). Model compression techniques such as pruning (Gale et al., 2019), knowledge distillation (Hinton et al., 2015), or quantization (Gholami et al., 2021) are crucial towards enabling the deployment of large models across a wide range of platforms, including resource-constrained edge devices like smartphones.

Despite achieving comparable performance at an aggregate level over the entire dataset, pruned models often exhibit significant accuracy reduction for some data sub-groups (Hooker et al., 2019; 2020; Paganini, 2020). In particular, under-represented groups can suffer high performance degradation while the overall performance remains unaffected, thus exacerbating systemic biases in machine learning models. Tran et al. (2022) refer to this phenomenon as the *disparate impact of pruning*.

Existing mitigation methods face challenges in terms of interpretability and scalability to a large number of sub-groups. Tran et al. (2022) introduce constraints aiming to equalize the loss of the sparse model across sub-groups. However, their approach does not account for the unequal group-level performance of the dense model. Moreover, while the loss can be a useful surrogate for training, this method addresses the disparate impact issue indirectly as it focuses on controlling the loss, rather than group-level changes in accuracy. Alternatively, Lin et al. (2022) compute per-group importance scores for every model parameter to determine the weights to be pruned. This approach becomes prohibitively expensive when the model or the number of sub-groups is large.

In this work, we characterize the disparate impact of pruning in terms of the group-level accuracy gaps between the dense and sparse models. Additionally, we propose a problem *formulation that directly addresses the disparate impact of pruning* by imposing constraints on the per-group excess accuracy gaps (CEAG). A key advantage of our proposed formulation is that it *enjoys interpretable semantics*: feasible solutions of our optimization problem correspond to models with low pruning-induced disparity. Finally, our approach introduces a *negligible computational overhead* (Appendix E.1) compared to (disparity-agnostic) naive fine-tuning of the sparse model, making it applicable to problems with large numbers of groups, such as intersectional fairness tasks.

---

*Equal contribution. Contact: merajhashemi@yahoo.co.uk, juan.ramirez@mila.quebec.

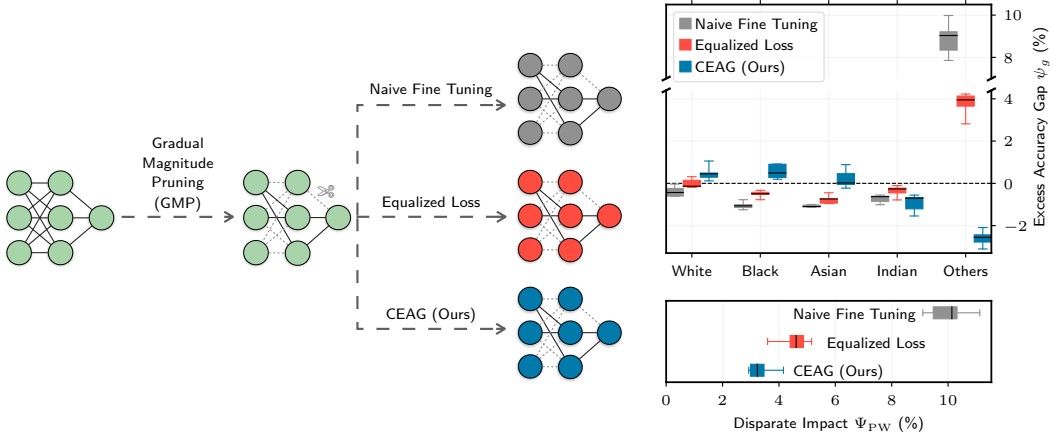

Figure 1: **Left:** A dense model is sparsified with GMP, and then subjected to either (i) naive fine-tuning (NFT, using ERM), (ii) equalized loss constraints (Tran et al., 2022, EL), or (iii) our approach (CEAG). **Right:** Positive (resp. negative) excess accuracy gaps (EAGs, §3.1) indicate groups whose performance degraded more (resp. less) than the model's overall accuracy change. Models with low disparate impact have EAGs that concentrate around zero. **CEAG consistenly yields models with lower disparity ($\Psi_{PW}$, §3.1) than NFT and EL.** For example, NFT yields a 10% hyper-degradation (EAG, $\psi_g$) on group *Others*. Results correspond to race prediction on UTKFace, with race as group attribute at 90% sparsity. Metrics are measured on the training set and averaged over 5 seeds.

Fig. 1 illustrates the reliability of our approach at mitigating the disparate impact of pruning. We measure disparity in terms of excess accuracy gaps (EAGs, §3.1). Naive fine-tuning yields models that disproportionately affect group *Others*, and while the equalized loss formulation mitigates the issue, *our formulation consistently reduces the pruning-induced disparity*. See §5 for further discussion.

The main contributions of our work[1] are as follows:

- We formulate a constrained optimization problem (CEAG, §3) that directly controls disparate impact by bounding group-level accuracy gaps between the dense and sparse models.

- We propose an algorithm for solving constrained optimization problems with *non-differentiable, stochastic constraints* (§4). We use proxy constraints (Cotter et al., 2019) to address non-differentiablity; and introduce replay buffers (§4.2) for handling noise in the estimation of constraints.

- Our replay buffers improve the training dynamics of the equalized loss formulation proposed by Tran et al. (2022). The improved dynamics lead to better models in terms of disparity.

- Our experiments demonstrate that we can reliably mitigate the disparate impact of pruning across multiple architectures, datasets, and sparsity levels (§5). These results carry over to tasks with intersectional groups, and up to hundreds of constraints.

Our experimental results indicate that *all methods considered in this paper (including ours) fail to mitigate pruning-induced disparities on unseen data*. To the best of our knowledge, we are the first to document this generalization challenge. Despite this, our proposed method constitutes a step in the right direction since our approach is *the only one* that reliably mitigates the disparate impact of pruning on the training set. We hope our empirical observations will motivate further research on improving the generalization properties of methods for mitigating the disparate impact of pruning.

## 2 RELATED WORKS

**Disparate Impact of Pruning.** Hooker et al. (2019; 2020) and Paganini (2020) document the disparate impact of pruning where some classes experience a more significant performance degradation compared to others. Existing methods to mitigate disparity involve fairness-aware pruning (Lin et al., 2022) or formulating constraints on a surrogate metric such as the loss (Tran et al., 2022).

---

[1]Our code is available here: https://github.com/merajhashemi/balancing-act

Lin et al. (2022) propose a pruning technique that removes weights based on a heuristic metric that relates parameters with their importance for predicting samples from each group. This approach scales poorly as it requires computing importance scores for each weight *and* group.

Tran et al. (2022) apply constraints to match the sparse model's *loss* on each sub-group to the aggregate loss. These constraints are (i) agnostic to the performance of the *dense* model on each group and (ii) are based on the loss, which is a surrogate metric for assessing the accuracy-based disparate impact. Since the disparate impact of pruning is measured with respect to a reference model, the equalized loss formulation addresses the problem indirectly. Moreover, loss-based constraints lack the interpretability of the per-group accuracy changes between the sparse and dense models.

**Fairness and Constraints.** Independent of model pruning, fairness in machine learning models is a well studied problem (Dwork et al., 2012; Dieterich et al., 2016; Verma & Rubin, 2018; Mehrabi et al., 2021; Zemel et al., 2013; Zhao & Gordon, 2022). Enforcing fairness with constraints has mainly focused on imposing requirements such as demographic parity, equalized odds, equal opportunity (Hardt et al., 2016), accuracy parity (Agarwal et al., 2018; Berk et al., 2021), or combinations of these properties (Zafar et al., 2017; Lowy et al., 2021; Bakker et al., 2020; Shui et al., 2022). The *disparate impact of pruning* is a fairness notion in the context of sparsity that aims to match the performance of a sparse model to that of a reference dense model.

**Constrained Optimization.** Constrained formulations have gained popularity in different sub-fields of machine learning such as safe reinforcement learning (Stooke et al., 2020), active learning (Elenter et al., 2022) and sparsity (Gallego-Posada et al., 2022). These constrained formulations lead to stochastic min-max optimization problems, which can be challenging to optimize due to their non-convexity (Lin et al., 2020). We make use of proxy constraints (Cotter et al., 2019) to solve problems with interpretable but non-differentiable constraints.

**Variance Reduction.** The stochasticity in gradient estimates introduces additional optimization challenges (Beznosikov et al., 2023). Variance reduction techniques (Gower et al., 2020) have been employed to improve convergence on stochastic optimization (Defazio et al., 2014), and in min-max games (Chavdarova et al., 2019). In this work, we leverage the idea of replay buffers (Mnih et al., 2013) to reduce the noise in the estimation of stochastic constraints.

## 3 ADDRESSING THE DISPARATE IMPACT OF PRUNING VIA ACCURACY GAPS

In this section, we propose using *accuracy gaps* (AGs) to quantify the disparate impact induced by model pruning. AGs are group-level measurements that quantify changes in accuracy between the dense and sparse models. As we will see, large discrepancies in AGs across groups correspond to scenarios where pruning-induced disparity is high. In §3.2, we propose a problem formulation that yields models with low disparity by explicitly constraining deviations in the group accuracy gaps.

### 3.1 ACCURACY GAPS

We consider a supervised learning problem on a dataset $\mathfrak{D} = \{(\boldsymbol{x}_i, y_i, g_i)\}_{i=1}^N$ of $N$ i.i.d tuples, each comprising features $\boldsymbol{x} \in \mathcal{X}$, target class $y \in [K]$ and group membership $g \in \mathcal{G}$. The dataset can be partitioned into *sub-groups* $\mathfrak{D}_g \triangleq \{(\boldsymbol{x}_i, y_i, g_i) \in \mathfrak{D} \mid g_i = g\}$ for every $g \in \mathcal{G}$.

Let $h_{\boldsymbol{\theta}} : \mathcal{X} \to \mathbb{R}^K$ be a predictor with parameters $\boldsymbol{\theta} \in \Theta$. The accuracy of $h_{\boldsymbol{\theta}}$ on a sample set $\mathcal{D}$ is $A(\boldsymbol{\theta}|\mathcal{D}) \triangleq \frac{1}{|\mathcal{D}|} \sum_{(\boldsymbol{x},y,g) \in \mathcal{D}} \mathbb{1}\{\text{argmax}[h_{\boldsymbol{\theta}}(\boldsymbol{x})] = y\}$. In particular, $A(\boldsymbol{\theta}|\mathfrak{D})$ denotes the model accuracy on the entire dataset, while $A(\boldsymbol{\theta}|\mathfrak{D}_g)$ is the model accuracy on a specific sub-group $g$.

Given access to a dense pre-trained model, we are interested in the effect of pruning on the accuracy across sub-groups $\mathfrak{D}_g$. In realistic pruning applications the dense model may exhibit different accuracies across sub-groups, thus we do not aim to equalize the accuracy of the sparse model across groups. Therefore, **we argue that the accuracies after pruning should change (approximately) equally across sub-groups.**

Let $\boldsymbol{\theta}_d$ and $\boldsymbol{\theta}_s$ denote the parameters of the dense and sparse models, respectively. We define the *global accuracy gap* $\Delta(\boldsymbol{\theta}_s, \boldsymbol{\theta}_d)$ and *group accuracy gaps* $\Delta_g(\boldsymbol{\theta}_s, \boldsymbol{\theta}_d)$ as:

$$\Delta(\boldsymbol{\theta}_s, \boldsymbol{\theta}_d) \triangleq A(\boldsymbol{\theta}_d|\mathfrak{D}) - A(\boldsymbol{\theta}_s|\mathfrak{D}), \tag{1}$$

$$\Delta_g(\boldsymbol{\theta}_s, \boldsymbol{\theta}_d) \triangleq A(\boldsymbol{\theta}_d|\mathfrak{D}_g) - A(\boldsymbol{\theta}_s|\mathfrak{D}_g) \qquad \forall g \in \mathcal{G}. \tag{2}$$

A *positive gap* (resp. *negative*) corresponds to a *degradation* (resp. *improvement*) in the performance of the sparse model with respect to that of the dense model. This correspondence holds both at the global $\Delta\left(\boldsymbol{\theta}_s, \boldsymbol{\theta}_d\right)$ and group levels $\Delta_g\left(\boldsymbol{\theta}_s, \boldsymbol{\theta}_d\right)$.

**Disparate Impact of Pruning.** Following our discussion above, we say a sparse model $h_{\boldsymbol{\theta}_s}$ experiences low disparate impact (with respect to a dense model $h_{\boldsymbol{\theta}_d}$) if the changes in performance are similar across sub-groups, i.e. $\Delta_g\left(\boldsymbol{\theta}_s, \boldsymbol{\theta}_d\right) \approx \Delta_{g'}\left(\boldsymbol{\theta}_s, \boldsymbol{\theta}_d\right), \forall g, g' \in \mathcal{G}$.

Due to the loss of model capacity caused by pruning, typically $\Delta\left(\boldsymbol{\theta}_s, \boldsymbol{\theta}_d\right) > 0$. Thus, we consider $\Delta\left(\boldsymbol{\theta}_s, \boldsymbol{\theta}_d\right)$ as the reference point for defining the group *excess accuracy gaps* (EAGs):

$$\psi_g\left(\boldsymbol{\theta}_s, \boldsymbol{\theta}_d\right) \triangleq \Delta_g\left(\boldsymbol{\theta}_s, \boldsymbol{\theta}_d\right) - \Delta\left(\boldsymbol{\theta}_s, \boldsymbol{\theta}_d\right), \qquad \forall g \in \mathcal{G}. \tag{3}$$

If $\psi_g\left(\boldsymbol{\theta}_s, \boldsymbol{\theta}_d\right) > 0$, then $g$ is more negatively impacted by pruning than the overall dataset. Conversely, $\psi_{g'}\left(\boldsymbol{\theta}_s, \boldsymbol{\theta}_d\right) < 0$ indicates that group $g'$ was less affected relative to the overall model degradation.

Note that if $\psi_g = 0, \forall g \in \mathcal{G}$, then it follows that $\Delta_g\left(\boldsymbol{\theta}_s, \boldsymbol{\theta}_d\right) = \Delta_{g'}\left(\boldsymbol{\theta}_s, \boldsymbol{\theta}_d\right), \forall g, g' \in \mathcal{G}$, and there is no disparate impact. Thus, we quantify the disparate impact of pruning via:

$$\Psi_{\text{PairWise}}\left(\boldsymbol{\theta}_s, \boldsymbol{\theta}_d\right) \triangleq \max_{g, g' \in \mathcal{G}} \psi_g\left(\boldsymbol{\theta}_s, \boldsymbol{\theta}_d\right) - \psi_{g'}\left(\boldsymbol{\theta}_s, \boldsymbol{\theta}_d\right) = \max_{g \in \mathcal{G}} \Delta_g\left(\boldsymbol{\theta}_s, \boldsymbol{\theta}_d\right) - \min_{g' \in \mathcal{G}} \Delta_{g'}\left(\boldsymbol{\theta}_s, \boldsymbol{\theta}_d\right). \tag{4}$$

Note that $\Psi_{\text{PW}} \geq 0$ always. Moreover, $\Psi_{\text{PW}} = 0$ if and only if we are in an ideal setting where the accuracy gaps are *equal* across all groups. However, aiming to constraint $\Psi_{\text{PW}}$ directly can be difficult in practice (see Appendix B.3). Instead, we consider constraints on each individual group EAG.

## 3.2 CONSTRAINED EXCESS ACCURACY GAPS FORMULATION

We propose to impose upper-bounds (with a tolerance level $\epsilon \geq 0$) on the values of $\psi_g\left(\boldsymbol{\theta}_s, \boldsymbol{\theta}_d\right) \leq \epsilon$. Since $\epsilon \geq 0$, the constraints are effectively only enforced on $\psi_g\left(\boldsymbol{\theta}_s, \boldsymbol{\theta}_d\right) > 0$, corresponding to groups experiencing hyper-degradation in performance (with respect to the average degradation)[2]. Imposing a lower bound on group EAGs $\psi_g$ would allow for better control over the resulting disparate impact $\Psi_{\text{PW}}$. However, solving the problem with both of these bounds is challenging due to the small size of the feasible region relative to the estimation noise in the constraints. Appendix B.3 provides further discussion and motivation regarding the choice to constrain only positive $\psi_g$ values.

This choice motivates an *operational definition of disparate impact* which focuses on the group with the highest EAG, given by $\max_g \psi_g$. Bounding this quantity can be achieved by imposing constraints on every EAG. This gives rise to the following optimization problem with per-group constraints:

$$(\textsf{CEAG}) \quad \underset{\boldsymbol{\theta}_s \in \Theta}{\text{argmin}}\ L(\boldsymbol{\theta}_s | \mathfrak{D}), \quad \text{s.t.} \quad \psi_g\left(\boldsymbol{\theta}_s, \boldsymbol{\theta}_d\right) = \Delta_g\left(\boldsymbol{\theta}_s, \boldsymbol{\theta}_d\right) - \Delta\left(\boldsymbol{\theta}_s, \boldsymbol{\theta}_d\right) \leq \epsilon, \quad \forall g \in \mathcal{G} \tag{5}$$

where $L(\boldsymbol{\theta} | \mathcal{D})$ is the loss of $h_{\boldsymbol{\theta}}$ on dataset $\mathcal{D}$, and the tolerance $\epsilon \geq 0$ is the maximum allowed EAG.

When $\Delta\left(\boldsymbol{\theta}_s, \boldsymbol{\theta}_d\right) > 0$, the constraints require that the performance degradation for each group be at most the overall model degradation plus the tolerance. Conversely, if $\Delta\left(\boldsymbol{\theta}_s, \boldsymbol{\theta}_d\right) < 0$, the constraints prescribe that all group accuracies must *increase* by at least the overall improvement, except for an $\epsilon$.

## 3.3 DISCUSSION

By formulating constraints on EAGs, CEAG directly addresses the disparate impact of pruning and has benefits in terms of interpretability, flexibility, and accountability. See Appendix B for alternative constrained formulations for addressing the disparate impact of pruning.

**Tackling disparate impact.** Existing methods aim to mitigate disparate impact by enforcing properties on the sparse model while being agnostic to the performance of the dense model. Since EAGs relate the per-group performance of the dense and sparse models, we argue that our approach *actually* addresses pruning-induced disparity, rather than other fairness notions such as loss equalization as proposed by Tran et al. (2022).

**Interpretability.** The choice of tolerance level $\epsilon$ directly translates to bounds on AGs. For example, setting $\epsilon = 1\%$ implies the worst affected class may not lose beyond $1\%$ accuracy compared to the

---

[2]Note that the set of hyper-degraded groups $\{g \in \mathcal{G} \mid \psi_g\left(\boldsymbol{\theta}_s, \boldsymbol{\theta}_d\right) > 0\}$ depends directly on the parameters of the sparse model $\boldsymbol{\theta}_s$ and thus changes at every training step.

overall model change. In contrast, it is challenging to set interpretable tolerance levels for constraints based on losses.

**Flexibility.** CEAG allows for some slack in the disparity of the pruned model, as prescribed by the tolerance $\epsilon$. This flexibility allows incorporating application-specific requirements into the learning procedure. For example, small tolerance values allow enforcing strict fairness regulations. Moreover, this flexibility may be necessary in practice since the reduced capacity of the sparse model can make it impossible to attain $\Delta_g(\boldsymbol{\theta}_s, \boldsymbol{\theta}_d) = \Delta(\boldsymbol{\theta}_s, \boldsymbol{\theta}_d) \; \forall g \in \mathcal{G}$.

**Accountability.** Being a constrained approach, establishing feasibility with respect to CEAG constitutes a clear success criterion to determine if a pruned model achieves acceptable disparity levels: a model is only admissible if it satisfies the constraints at a prescribed tolerance level.

# 4 SOLVING THE CONSTRAINED EXCESS ACCURACY GAPS PROBLEM

A popular approach to solve constrained optimization problems such as CEAGin Eq. (5) is to formulate its Lagrangian and optimize the resulting min-max problem:

$$\min_{\boldsymbol{\theta}_s \in \Theta} \max_{\boldsymbol{\lambda} \geq 0} \mathfrak{L}(\boldsymbol{\theta}_s, \boldsymbol{\lambda}) \triangleq L(\boldsymbol{\theta}_s|\mathfrak{D}) + \sum_{g \in \mathcal{G}} \lambda_g \left( \psi_g(\boldsymbol{\theta}_s, \boldsymbol{\theta}_d) - \epsilon \right), \tag{6}$$

where $\lambda_g \geq 0$ is the Lagrange multiplier associated with the constraint for group $g$ and $\boldsymbol{\lambda} = [\lambda_g]_{g \in \mathcal{G}}$. We refer to $\boldsymbol{\theta}_s$ as the *primal* parameters, and to $\boldsymbol{\lambda}$ as the *dual* parameters.

Optimizing deep neural networks can be challenging, and generally requires carefully crafted procedures and extensive hyper-parameter tuning (Choi et al., 2019). We are interested in re-using standard techniques for optimizing $\boldsymbol{\theta}_s$. Therefore, we consider a generic optimization protocol on $\boldsymbol{\theta}_s$ and gradient ascent on $\boldsymbol{\lambda}$, instead of specialized optimization approaches for min-max games such as extragradient (Gidel et al., 2019; Korpelevich, 1976).

## 4.1 OPTIMIZATION WITH NON-DIFFERENTIABLE CONSTRAINTS

A natural next step is to optimize Eq. (6) with gradient-based updates. Unfortunately, this is not possible as the $\psi_g$ terms are not continuous (since they are accuracy gaps), and are non-differentiable with respect to $\boldsymbol{\theta}_s$. Therefore, we must resort to a surrogate $\tilde{\psi}_g$ for computing gradients with respect to $\boldsymbol{\theta}_s$. In contrast, Eq. (6) is differentiable with respect to $\boldsymbol{\lambda}$, with gradients corresponding to constraint violations. Thus, the dual variables can be updated using the non-differentiable constraints $\psi_g$. This update scheme is inspired by the proxy-constraint technique introduced by Cotter et al. (2019).

$$\boldsymbol{\theta}_s^*, \boldsymbol{\lambda}^* \in \begin{cases} \underset{\boldsymbol{\theta}_s \in \Theta}{\operatorname{argmin}} \; \mathfrak{L}_\theta(\boldsymbol{\theta}_s, \boldsymbol{\lambda}) \triangleq L(\boldsymbol{\theta}_s|\mathfrak{D}) + \sum_{g \in \mathcal{G}} \lambda_g \tilde{\psi}_g(\boldsymbol{\theta}_s, \boldsymbol{\theta}_d) \\ \underset{\boldsymbol{\lambda} \geq 0}{\operatorname{argmax}} \; \mathfrak{L}_\lambda(\boldsymbol{\theta}_s, \boldsymbol{\lambda}) \triangleq \sum_{g \in \mathcal{G}} \lambda_g \left( \psi_g(\boldsymbol{\theta}_s, \boldsymbol{\theta}_d) - \epsilon \right), \end{cases} \tag{7}$$

Specifically, we choose surrogates $\tilde{\psi}_g$ given by the *excess (negative) loss gaps*: $\tilde{\psi}_g(\boldsymbol{\theta}_s, \boldsymbol{\theta}_d) \triangleq -\left( L(\boldsymbol{\theta}_d|\mathfrak{D}_g) - L(\boldsymbol{\theta}_s|\mathfrak{D}_g) \right) + \left( L(\boldsymbol{\theta}_d|\mathfrak{D}) - L(\boldsymbol{\theta}_s|\mathfrak{D}) \right)$. Note that $\tilde{\psi}_g$ has the same structure as $\psi_g$, but replaces accuracy measurements with *negative* loss terms. This is a reasonable choice of surrogate function since *drops* in accuracy for the sparse model correspond to *increases* in loss.

Eq. (7) represents a two-player, non-zero-sum game. Rather than replacing the non-differentiable constraints with their surrogates everywhere, this approach only performs the replacement *when necessary*, i.e., for computing gradients for the primal parameters. Preserving the actual constraints on the dual objective $\mathfrak{L}_\lambda(\boldsymbol{\theta}_s, \boldsymbol{\lambda})$ is useful as it results in a problem closer to Eq. (6).

Equation (7) can be optimized via gradient descent on $\boldsymbol{\theta}_s$ (based on $\mathfrak{L}_\theta$) and gradient ascent on $\boldsymbol{\lambda}$ (based on $\mathfrak{L}_\lambda$). Alternating gradient descent-ascent (Alt-GDA) updates yield:

$$\lambda_g^{(t+1)} = \left[ \lambda_g^{(t)} + \eta_\lambda \left( \psi_g\left( \boldsymbol{\theta}_s^{(t)}, \boldsymbol{\theta}_d \right) - \epsilon \right) \right]_+ \tag{8}$$

$$\boldsymbol{\theta}_s^{(t+1)} = \boldsymbol{\theta}_s^{(t)} - \eta_\theta \left[ \nabla_\theta L\left( \boldsymbol{\theta}_s^{(t)}|\mathfrak{D} \right) + \sum_{g \in \mathcal{G}} \lambda_g^{(t+1)} \nabla_\theta \tilde{\psi}_g\left( \boldsymbol{\theta}_s^{(t)}, \boldsymbol{\theta}_d \right) \right], \tag{9}$$

where $\eta_\theta$ and $\eta_\lambda$ are step-sizes and $[\cdot]_+ = \max(\cdot, 0)$. We initialize the Lagrange multipliers to $\boldsymbol{\lambda}^{(0)} = \mathbf{0}$. Appendix A contains more details on non-convex constrained optimization.

---

**Algorithm 1** Constrained Excess Accuracy Gap (CEAG)

---

**Input:** $\theta$: Initial model parameters, $\eta_\theta$: Primal step-size, $\eta_\lambda$: Dual step-size, $k$: Memory size for replay buffer, $\epsilon$: Tolerance hyper-parameter, $B$: Batch size, $T$: Total number of iterations, $A_{\text{dense}}^g$: Accuracy of the dense model on each group $g$, $A_{\text{dense}}$: Aggregate accuracy of the dense model.

1: $\lambda_g \leftarrow 0, \quad \forall g \in \mathcal{G}$                                   ▷ *Initialize dual parameters*
2: $\texttt{buf}_g \leftarrow \texttt{queue}(k), \quad \forall g \in \mathcal{G}$                      ▷ *Initialize replay buffer*
3: **for** $\texttt{iter} = 1, \ldots, T$ **do**
4:      $\mathbf{x}, \mathbf{y}, \mathbf{g} \leftarrow \text{Sample } \{(x_i, y_i, g_i)\}_{i=1}^B \sim \mathfrak{D}$           ▷ *Sample batch from training set*
5:      $\texttt{idx}_g \leftarrow (\mathbf{g} == g), \quad \forall g \in \mathcal{G}$         ▷ *Calculate sub-group indices for batch*
6:      $\hat{\mathbf{y}} \leftarrow h_\theta(\mathbf{x})$                                           ▷ *Compute forward-pass*
7:      $\texttt{buf}_g \leftarrow \text{UPDATEBUFFER}(\texttt{buf}_g, \hat{\mathbf{y}}, \mathbf{y}, \texttt{idx}_g), \quad \forall g \in \mathcal{G}$     ▷ *Update replay buffer*
8:      $\psi_g \leftarrow \text{QUERYBUFFERS}(\{\texttt{buf}_g\}_{g=1}^{\mathcal{G}}, k, \{A_{\text{dense}}^g\}_{g=1}^{\mathcal{G}}, A_{\text{dense}})$    ▷ *Query replay buffers*
9:      $\tilde{\psi}_g \leftarrow \text{COMPUTESURROGATE}(\hat{\mathbf{y}}, \mathbf{y}, \texttt{idx}_g), \quad \forall g \in \mathcal{G}$       ▷ *Compute surrogates*
10:     $\lambda_g \leftarrow \max\{0, \lambda_g + \eta_\lambda(\psi_g - \epsilon)\}, \quad \forall g \in \mathcal{G}$       ▷ *Update dual params*
11:     $\texttt{grad}_\theta \leftarrow \nabla_\theta \left[ L\left(\theta | (\mathbf{x}, \mathbf{y})\right) + \sum_{g \in \mathcal{G}} \lambda_g \tilde{\psi}_g \right]$        ▷ *Compute primal gradient*
12:     $\theta \leftarrow \text{PRIMALOPTIMUPDATE}(\eta_\theta, \texttt{grad}_\theta)$             ▷ *Update model params*
13: **end for**
14: **return** $\theta$

---

## 4.2 STOCHASTIC CONSTRAINTS AND REPLAY BUFFERS

In practice, the problem in Eq. (5) is solved by using mini-batch samples from the dataset to estimate the objective function, the constraints, and their gradients. This procedure can yield constraint estimates with high variance across mini-batches, especially for under-represented groups; or for all groups when the number of constraints is large. In extreme cases, a mini-batch may contain very few samples from a given sub-group, leading to multiplier updates based on very noisy estimates.

We overcome these issues by estimating constraints based on information across multiple mini-batches. For calculating AGs, (i) we compute the performance of the dense model *on the whole dataset* (once at the beginning of training), and (ii) we estimate the accuracy of the sparse model from per-sample accuracy measurements on the $k$ most recent datapoints of each group. We refer to the data structure that stores historic accuracies as a *replay buffer* (RB), given the analogy to the technique used in reinforcement learning (Mnih et al., 2013). The choice of buffer size $k$ introduces a trade-off between reducing the variance of the constraints, and biasing estimates towards old measurements.

These adjustments reduce variance in the estimation of the constraints, thus yielding stable updates for the multipliers. This allows us to solve Eq. (5) in settings with large numbers of constraints relative to the choice of batch size. We do not apply variance reduction on the model updates. For details on our implementation of replay buffers, see Appendix C. For experimental evidence on their benefits, see §5.3 and Appendix C.1.

## 4.3 ALGORITHMIC DETAILS

Algorithm 1 presents our approach for solving CEAG. Note that Algorithm 1 is applicable to a broader class of constrained optimization problems with stochastic constraints, including the equalized loss formulation of Tran et al. (2022) (see Appendix B.1 for details).

**Computational Overhead.** The constrained approach in Algorithm 1 represents a negligible computational overhead compared to fine-tuning the sparse model with empirical risk minimization. An iteration of Alt-GDA (Eq. (8)) requires *one forward pass and one backward pass* through the model since the same iterate of $\boldsymbol{\theta}_s$ is used for both the primal and dual updates. This matches the cost of gradient descent for ERM, except for the minimal overhead associated with the evaluation of constraints after the forward pass. Note that, given our choice of surrogate, the gradient of the Lagrangian with respect to $\boldsymbol{\theta}_s$ is a weighted average of the per-sample loss gradients, which autograd

frameworks can compute as efficiently as $\nabla_\theta L\left(\boldsymbol{\theta}_s | \mathfrak{D}\right)$. For empirical evidence supporting the claim that CEAG has negligible computational overhead compared to ERM, see Appendix E.1.

**Memory Cost.** The memory overhead of our approach is negligible in the context of training deep networks: storing the dual variables requires one float per constraint, and the replay buffers store only $|\mathcal{G}|$ booleans for each one of the $k$ slots in the buffer memory.

## 5  EXPERIMENTS

In this section, we present an empirical comparison between naive fine-tuning, equalized loss (Tran et al., 2022), and our proposed CEAG approach. The main goal of our experiments is to train sparse models with low pruning-induced disparity. While low disparity may introduce a trade-off with aggregate performance, we aim to achieve comparable overall accuracy to mitigation-agnostic methods. We explore the reliability and accountability of our approach, along with the effect of replay buffers on the constrained optimization problem. Our experiments demonstrate that our method successfully scales to problems with hundreds of groups.

### 5.1  EXPERIMENTAL SETUP

**Tasks and architectures.** We carry out experiments on the FairFace (Kärkkäinen & Joo, 2021) and UTKFace (Zhang et al., 2017) datasets, following the works of Lin et al. (2022) and Tran et al. (2022). Additionally, we perform experiments on CIFAR-100 (Krizhevsky, 2009), a task with a large number of sub-groups. The choice of target and group attributes for each dataset is specified in Appendix D.1. The architectures for each task, and the source of our pre-trained models are presented in Appendices D.3 and D.4, respectively.

**Baseline methods.** We compare with three baseline mitigation methods (i) NFT: the last iterate when fine-tuning the sparse model via ERM, (ii) NFT+ES: the best iterate of NFT in terms of test accuracy (early stopping), and (iii) EL+RB: our re-implementation of the equalized loss formulation proposed by Tran et al. (2022), enhanced with replay buffers (see Appendix B.1). The optimization hyper-parameters employed for each mitigation method (including CEAG) are described in Appendix D.6.

**Model pruning.** Previous work has shown that gradual magnitude pruning (GMP) (Zhu & Gupta, 2017) achieves SOTA aggregate performance on unstructured sparsity tasks (Blalock et al., 2020). Because of this (and its simplicity), we employ unstructured GMP on all our tasks. GMP gradually prunes the model by removing parameters with the smallest magnitude once every epoch. The remaining weights are fine-tuned in between pruning episodes. We carry out GMP during the first 15 epochs. Appendix D.5 provides further details on our pruning protocol.

**Choice of sparsity levels.** For very high levels of unstructured sparsity (over 95%), Gale et al. (2019) observe that pruning has a devastating impact on the overall performance of ResNet-50 models (He et al., 2016). In contrast, performance remains essentially unaffected for models with up to 85% sparsity. These observations may not carry over to other architectures such as MobileNets (Sandler et al., 2018), or other ResNets. Nonetheless, our experiments stick to the [85%, 95%] range, except for FairFace experiments, where we consider 99% sparsity, akin to FairGrape (Lin et al., 2022).

**Software.** Our implementations use PyTorch 1.13.0 (Paszke et al., 2019) and the Cooper library for constrained optimization (Gallego-Posada & Ramirez, 2022).

**Experimental uncertainty.** All metrics reported in our tables and plots follow the pattern `avg ±` `std`. Unless mentioned otherwise, all our experimental metrics are aggregated across 5 seeds.

For comprehensive experimental results across multiple tasks and sparsity levels, see Appendix F.

### 5.2  FAIRFACE AND UTKFACE

**ResNet-34 Models on FairFace.** Table 1 includes results for FairFace classification at 99% sparsity. We compare the behavior of NFT, NFT+ES, EL+RB, and CEAG. We quote the results reported for the FairGRAPE technique[3], aggregated over 3 seeds.

We observe that CEAG attains a feasible model in training ($\max_g \psi_g \leq \epsilon$), as well as the smallest $\max_g \psi_g$ both in the training and test sets. This does not come at the cost of aggregate performance,

---

[3]We do not re-run FairGRAPE owing to its high computational cost, see discussion in Appendix E.2

as all methods achieve a comparable test accuracy of around 65%. We observe that FairGRAPE's $\max_g \psi_g$ and $\Psi_{\mathrm{PW}}$ are significantly higher than that of all other methods.

Table 1: Race prediction task on FairFace with race as group attribute. **CEAG achieves a** $\max_g \psi_g$ **within the prescribed threshold**. Tol ($\epsilon$) is the tolerance hyper-parameter of CEAG. We do not specify $\epsilon$ for other formulations as they do not admit a tolerance.

| Sparsity | Method | Train | | | | Test | | |
|---|---|---|---|---|---|---|---|---|
| | | Accuracy | $\Psi_{\mathrm{PW}}$ | $\max_g \psi_g$ | Tol ($\epsilon$) | Accuracy | $\Psi_{\mathrm{PW}}$ | $\max_g \psi_g$ |
| 99 | NFT | $76.1 \pm 0.2$ | $3.9 \pm 0.9$ | $2.3 \pm 0.3$ | – | $65.2 \pm 0.4$ | $4.2 \pm 0.5$ | $2.1 \pm 0.5$ |
| | NFT + ES | $74.0 \pm 2.5$ | $7.2 \pm 3.3$ | $4.0 \pm 1.4$ | – | $65.4 \pm 0.4$ | $6.3 \pm 2.6$ | $2.9 \pm 1.3$ |
| | EL + RB | $76.1 \pm 0.1$ | $8.8 \pm 1.3$ | $2.6 \pm 0.2$ | – | $65.1 \pm 0.4$ | $6.0 \pm 1.5$ | $2.4 \pm 0.4$ |
| | FairGRAPE | – | – | – | – | $65.1$ | $15.9$ | $10.7$ |
| | CEAG | $76.2 \pm 0.1$ | $3.5 \pm 0.6$ | $1.8 \pm 0.4$ | $\leq 2\%$ ✓ | $65.2 \pm 0.4$ | $4.3 \pm 0.8$ | $2.0 \pm 0.3$ |

Table 2: Race prediction task on the UTKFace dataset with the intersection of race and gender as group attribute. For instance, if a sample has race *Black* and gender *Female*, its group is *Black-Female*. **CEAG consistently achieves a** $\max_g \psi_g$ **within the threshold, across sparsities**.

| Sparsity | Method | Train | | | | Test | | |
|---|---|---|---|---|---|---|---|---|
| | | Accuracy | $\Psi_{\mathrm{PW}}$ | $\max_g \psi_g$ | Tol ($\epsilon$) | Accuracy | $\Psi_{\mathrm{PW}}$ | $\max_g \psi_g$ |
| 90 | NFT | $98.1 \pm 0.1$ | $11.5 \pm 0.7$ | $10.0 \pm 0.7$ | – | $79.6 \pm 0.5$ | $8.9 \pm 2.3$ | $3.1 \pm 0.5$ |
| | NFT + ES | $90.5 \pm 4.7$ | $49.8 \pm 23.0$ | $44.8 \pm 20.8$ | – | $81.0 \pm 0.2$ | $12.0 \pm 5.3$ | $6.9 \pm 4.8$ |
| | EL + RB | $98.3 \pm 0.2$ | $3.2 \pm 0.6$ | $2.4 \pm 0.6$ | – | $79.4 \pm 0.5$ | $11.4 \pm 0.9$ | $3.0 \pm 1.1$ |
| | CEAG | $96.2 \pm 0.1$ | $2.4 \pm 0.6$ | $1.0 \pm 0.3$ | $\leq 3\%$ ✓ | $80.2 \pm 0.1$ | $6.0 \pm 2.5$ | $2.3 \pm 1.0$ |
| 92.5 | NFT | $95.1 \pm 0.2$ | $34.2 \pm 1.6$ | $30.7 \pm 1.5$ | – | $79.2 \pm 0.2$ | $8.8 \pm 3.2$ | $3.6 \pm 1.3$ |
| | NFT + ES | $91.2 \pm 2.7$ | $53.3 \pm 9.6$ | $48.0 \pm 8.3$ | – | $80.4 \pm 0.4$ | $7.5 \pm 3.4$ | $5.4 \pm 3.1$ |
| | EL + RB | $95.4 \pm 0.3$ | $11.1 \pm 1.5$ | $8.6 \pm 1.4$ | – | $78.7 \pm 0.3$ | $16.3 \pm 3.9$ | $3.3 \pm 0.6$ |
| | CEAG | $93.4 \pm 0.3$ | $3.8 \pm 0.4$ | $2.3 \pm 0.4$ | $\leq 3\%$ ✓ | $79.5 \pm 0.1$ | $10.8 \pm 2.2$ | $3.3 \pm 1.0$ |

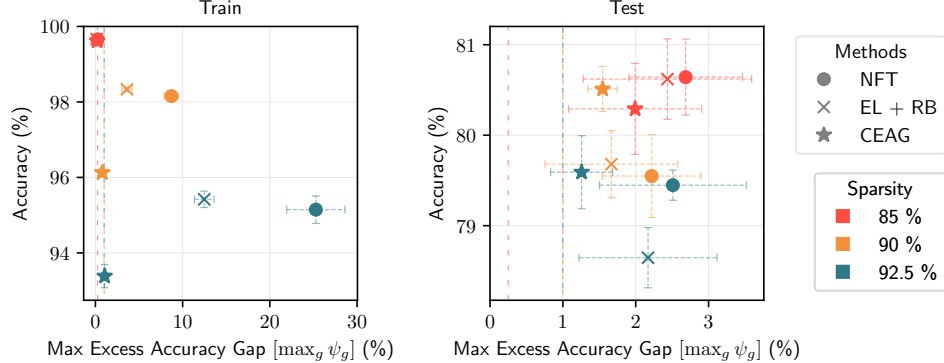

Figure 2: Trade-off between disparity and accuracy for UTKFace race prediction with race as group attribute. NFT and EL+RB yield models with high disparity. **In contrast, CEAG consistently produces models that mitigate the disparate impact of pruning.** CEAG's gains do not entail a degradation in overall test accuracy. Vertical dashed lines indicate the tolerance ($\epsilon$) of our method, with colors corresponding to different sparsity levels.

**MobileNet-V2 Models on UTKFace.** Fig. 2 illustrates results for UTKFace with race as group attribute. CEAG consistently attains feasible models in training, and the smallest values of $\max_g \psi_g$ in the test set. CEAG attains comparable performance to NFT and EL+RB in the test set.

Table 2 presents results for UTKFace with intersectional groups (race ∩ gender). NFT and NFT+ES have very high disparity metrics. In contrast, CEAG attains a feasible $\max_g \psi_g$ and the smallest $\Psi_{\mathrm{PW}}$ in the training set, for all sparsities. Our approach has worse aggregate performance than NFT and EL+RB in the train set; however, the test accuracy of these three methods is comparable.

For NFT, both Fig. 2 and Table 2 show significantly higher disparity metrics in training when compared to in test. This is an indicator that the sparse model achieves good performance in training by overfitting to the majority groups and losing a lot of performance on the under-represented groups.

### 5.3 SCALING TO LARGE NUMBERS OF GROUPS

**CifarResNet-56 models on CIFAR-100.** Table 3 contains results for CIFAR-100 classification at 92.5% sparsity. By having the groups correspond to class labels, constrained formulations for this experiment have 100 constraints. We include two additional experiments to illustrate the importance of replay buffers: equalized loss (EL), and CEAG (no RB), both *without replay buffers*.

Disparity metrics for EL and CEAG are better when employing replay buffers, both on the train and test sets. This difference is more notable for EL. We also observe the RBs improve the training dynamics of the dual variables (Appendix C.1). CEAG obtains the best disparity on the train set. Nonetheless, all approaches have a significant generalization gap in terms of disparity measurements. We observe that the best accuracy and the smallest $\max_g \psi_g$ on the test set are obtained by EL+RB.

Table 3: CIFAR-100 classification with the group attributes being the class labels, at 92.5% sparsity. EL is the equalized loss formulation without replay buffers; CEAG (no RB) is similarly defined.

| Sparsity | Method | Train | | | | Test | | |
|---|---|---|---|---|---|---|---|---|
| | | Accuracy | $\Psi_{\text{PW}}$ | $\max_g \psi_g$ | Tol ($\epsilon$) | Accuracy | $\Psi_{\text{PW}}$ | $\max_g \psi_g$ |
| 92.5 | NFT | $99.8 \pm 0.0$ | $3.7 \pm 0.9$ | $3.0 \pm 0.9$ | – | $64.9 \pm 0.4$ | $26.2 \pm 5.2$ | $14.3 \pm 3.4$ |
| | NFT + ES | $99.3 \pm 0.2$ | $6.8 \pm 1.9$ | $5.8 \pm 1.8$ | – | $65.2 \pm 0.4$ | $27.4 \pm 2.3$ | $14.6 \pm 2.0$ |
| | EL | $98.5 \pm 0.1$ | $11.3 \pm 0.9$ | $9.8 \pm 1.0$ | – | $65.3 \pm 0.5$ | $25.8 \pm 2.0$ | $14.1 \pm 1.3$ |
| | EL + RB | $99.5 \pm 0.0$ | $6.7 \pm 1.4$ | $5.7 \pm 1.5$ | – | $65.3 \pm 0.4$ | $24.2 \pm 2.9$ | $13.3 \pm 2.4$ |
| | CEAG (no RB) | $99.6 \pm 0.0$ | $2.6 \pm 0.3$ | $1.7 \pm 0.2$ | $\leq 2\%$ ✓ | $65.0 \pm 0.4$ | $27.2 \pm 2.6$ | $14.9 \pm 2.5$ |
| | CEAG | $99.6 \pm 0.0$ | $2.4 \pm 0.2$ | $1.6 \pm 0.1$ | $\leq 2\%$ ✓ | $64.8 \pm 0.3$ | $25.0 \pm 1.9$ | $13.8 \pm 1.2$ |

## 6 DISCUSSION

It is important to develop techniques that reliably mitigate the disparate impact of pruning since deploying pruned models can have downstream consequences. We observe that NFT is unsuccessful at doing this, and NFT+ES amplifies the disparity induced by pruning. In contrast, CEAG reduces disparity while achieving comparable aggregate performance to NFT. However, *we observe that all mitigation approaches may fail to mitigate disparate impact on unseen data*.

**Mitigating the disparate impact of pruning.** Unlike other mitigation methods, our approach consistently mitigates the disparate impact of pruning on the training set. We observe this across a wide range of tasks and architectures. In contrast, other mitigation approaches generally yield worse maximum degradation $\max_g \psi_g$. In particular, NFT+ES yields models with very high disparity.

**Accuracy trade-off.** CEAG may introduce a trade-off in terms of accuracy in order to satisfy the disparity requirements. On the train set, we observe a small degradation in performance in comparison to NFT, typically of at most 2%; on the test set, CEAG's accuracy is comparable to that of NFT.

**Reliability.** Our approach reliably yields models within the requested disparity levels. Moreover, CEAG results in the smallest variance of the $\max_g \psi_g$ and $\Psi_{\text{PW}}$ metrics across seeds.

**Generalization.** Although CEAG reliably satisfies the constraints on the train set, this may not transfer to the test set. We highlight that (i) these generalization issues are present for other mitigation methods, and (ii) our approach generally achieves better test disparity than the baselines. Improving the generalization of disparity mitigation methods is an important direction for future research.

## 7 CONCLUSION

In this paper, we explore mitigating the disparate impact of pruning. We formalize disparate impact in terms of accuracy gaps between the dense and sparse models, and propose a constrained optimization approach for mitigating it. Our formulation offers interpretable constraints and allows for algorithmic accountability. Although other methods can indirectly reduce disparity, our approach reliably addresses the disparate impact of pruning across a wide range of tasks, while attaining comparable aggregate performance. In particular, our method successfully scales to tasks with hundreds of subgroups. Despite the fact that current mitigation methods exhibit generalization issues, our approach represents a solid step towards mitigating the disparate impact of pruning.

ETHICS STATEMENT

- **Facial recognition.** Our paper makes use of datasets that contain face images. We focus on these datasets as they illustrate the disparate impact of pruning, and for comparisons with previous work. We would like to highlight that although our method focuses on reducing the disparate impact across groups, we do not endorse the use of our algorithm in facial recognition systems.

- **Data annotation.** We use the UTKFace (Zhang et al., 2017) and FairFace (Kärkkäinen & Joo, 2021) datasets in this work. These datasets include annotations for sensitive demographic attributes such as race, gender, and age. However, it is essential to recognize that these annotations represent normative ways of perceiving gender, race, and age, and we do not endorse or promote these normative categorizations.

- **Ethical sourcing of data.** We don't endorse using datasets where the data may not have been ethically sourced or the workers/subjects involved in the data collection process are not fairly compensated.

- **Fairness notions.** We explore a specific notion of fairness in this paper: the disparate impact of pruning. Our framework can be extended to other fairness notions by incorporating additional constraints. However, certain notions of fairness are incompatible with each other, and a "fair" model in one definition could be "unfair" with respect to another (Friedler et al., 2021). Therefore, our method should not be considered a solution to all notions of fairness.

- **Disparate impact of pruning.** In this paper, we propose a constrained optimization technique that mitigates the disparate impact of pruning and successfully solve the problem on the training data. Unfortunately, like all other surveyed techniques, we observe significant challenges at mitigating disparate impact on unseen data. We advise practitioners to consider the implications of these generalization challenges when deploying sparse models in real-world systems.

- **Deploying pruned models.** We hope our paper brings about an important discussion on the implications of deploying pruned deep learning models in edge devices. As shown in this work, despite the application of mitigation techniques, pruning can exacerbate systemic biases. In particular, given the generalization issues across mitigation methods, it could cause unintended consequences when used in commercial applications.

REPRODUCIBILITY STATEMENT

We provide our code[4], including scripts to replicate the experiments in this paper. The pseudo-code of our algorithm is described in Algorithm 1. Experimental details, as well as the hyper-parameters used in our experiments, are included in Appendix D. Our implementation uses the open-source libraries PyTorch (Paszke et al., 2019) and Cooper (Gallego-Posada & Ramirez, 2022).

ACKNOWLEDGEMENTS

This research was partially supported by an IVADO PhD Excellence Scholarship, the Canada CIFAR AI Chair program (Mila), a Google Excellence Award, and the Natural Sciences and Engineering Research Council of Canada (NSERC). Simon Lacoste-Julien is a CIFAR Associate Fellow in the Learning in Machines & Brains program.

This research was enabled in part by compute resources, software, and technical help by Mila.

We would like to thank Nazanin Sepahvand for support in the initial stages of the project. We would like to thank Christos Louizos for encouraging the pursuit of this research idea. Additionally, we would like to thank Marwa El-Halabi, Stefan Horoi and Albert Orozco Camacho for their feedback on the paper.

---

[4]Our code is available here: https://github.com/merajhashemi/balancing-act

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

# Appendix

## Table of Contents

## A    CONSTRAINED AND MIN-MAX OPTIMIZATION

Lagrangian-based constrained optimization has gained popularity in machine/deep learning owing to the fine-grained control it provides over specific properties of models (Cotter et al., 2019; Stooke et al., 2020; Elenter et al., 2022; Gallego-Posada et al., 2022; Hounie et al., 2023). In the context of our work, attaining a desired disparity level can be done directly by imposing the excess accuracy gap constraints presented in Eq. (5). In contrast, achieving bounded disparity by augmenting the training objective with *additive penalties* is challenging as it requires iteratively tuning a penalty coefficient per group (Gallego-Posada et al., 2022).

The Lagrangian-based approach involves solving a non-convex-concave min-max optimization problem. In general, as long as the constraints are differentiable, the min-max problem can be optimized with gradient-based updates. Fortunately, Cotter et al. (2019) show how even when the constraints are non-differentiable (but differentiable surrogates are available) *proxy constraints* can be used to find a semi-coarse correlated equilibrium of the min-max problem.

The solution to the min-max optimization problem (i.e. a saddle point) associated with the Lagrangian corresponds to a global constrained minimizer of the original constrained problem (Bertsekas, 1997). However, a saddle point of the Lagrangian may not exist for non-convex problems (Cotter et al., 2019; von Neumann, 1928).

In the context of machine learning, the success of adversarial formulations such as GANs (Goodfellow et al., 2014) and adversarial training (Madry et al., 2018) has sparked interest in min-max optimization. Lin et al. (2020) prove local linear convergence for simultaneous gradient descent-ascent in the non-convex-concave setting. Moreover, Zhang et al. (2022) prove local linear convergence of Alt-GDA in the strongly-convex-concave setting. They observe that the iteration complexity of Alt-GDA is optimal (Mokhtari et al., 2020), thus matching that of extragradient (Korpelevich, 1976; Gidel et al., 2019). These observations motivate our choice of Alt-GDA for optimizing Eq. (5).

Recent work has studied the statistical properties of constrained optimization problems (Chamon & Ribeiro, 2020; Chamon et al., 2022). This line of work has formulated PAC generalization bounds on feasibility and optimality, arguing that learning with constraints is *not* a more difficult problem than learning without constraints (Chamon & Ribeiro, 2020).

## B    ALTERNATIVE CONSTRAINED FORMULATIONS

This section elaborates on alternative constrained formulations for mitigating the disparate impact of pruning. Appendix B.1 presents the equalized loss formulation of Tran et al. (2022), Appendix B.2 describes a problem that constrains *excess loss gaps* of the sparse model, and Appendix B.3 formulates problems that (approximately) equalize the per-group excess accuracy gaps.

### B.1    EQUALIZED LOSS

Equation (10) presents the equalized loss formulation for mitigating disparate impact (Tran et al., 2022). This formulation matches the loss of each group with the overall loss.

$$\underset{\theta \in \Theta}{\operatorname{argmin}} \ L(\theta|\mathfrak{D}), \qquad \text{s.t.} \quad L(\theta|\mathfrak{D}_g) - L(\theta|\mathfrak{D}) = 0, \quad \forall g \in \mathcal{G} \tag{10}$$

Tran et al. (2022) provide theoretical arguments to link disparate impact (in terms of group-level excess loss gaps) to the loss on each group. This justifies their choice of constraints.

Our implementation of this approach follows a pipeline akin to Algorithm 1: we optimize it with alternating gradient descent-ascent and use group replay buffers to reduce variance in the estimation of the constraints for updating the dual variables. The storage cost associated with the buffer in this setting is higher than that for CEAG, since per-sample losses (floating point numbers) are stored instead of accuracies (booleans).

As shown in Appendix C.1, we notice smoother training dynamics for the multipliers when using the replay buffers. Table 3 shows how the equalized loss formulation benefits from them in terms of mitigating the disparate impact of pruning.

## B.2 Constrained Excess Loss Gaps

An alternative to both CEAG and Eq. (10) is to constrain loss gaps between the dense and sparse models. This yields the following *constrained excess loss gaps* problem:

$$\operatorname*{argmin}_{\boldsymbol{\theta}_s \in \Theta} L(\boldsymbol{\theta}_s | \mathfrak{D}) \tag{11}$$
$$\text{s.t.} \quad \tilde{\psi}_g = -\big(L(\boldsymbol{\theta}_d | \mathfrak{D}_g) - L(\boldsymbol{\theta}_s | \mathfrak{D}_g)\big) + \big(L(\boldsymbol{\theta}_d | \mathfrak{D}) - L(\boldsymbol{\theta}_s | \mathfrak{D})\big) \leq \epsilon, \quad \forall g \in \mathcal{G}$$

This formulation addresses the disparate impact of pruning, although in terms of loss gaps instead of accuracy gaps. Selecting a tolerance $\epsilon$ for this formulation can be challenging as it requires specifying acceptable levels of excess loss gaps, which can vary significantly across tasks.

## B.3 Constrained $\Psi_{\text{PW}}$

**Constrained disparate impact.** A natural formulation to consider involves constraining the disparate impact, as defined in Eq. (4). The constrained optimization can be formulated as:

$$\operatorname*{argmin}_{\boldsymbol{\theta}_s \in \Theta} L(\boldsymbol{\theta}_s | \mathfrak{D}), \qquad \text{s.t.} \quad \Psi_{\text{PW}} = \max_{g \in \mathcal{G}} \Delta_g (\boldsymbol{\theta}_s, \boldsymbol{\theta}_d) - \min_{g \in \mathcal{G}} \Delta_g (\boldsymbol{\theta}_s, \boldsymbol{\theta}_d) \leq \epsilon. \tag{12}$$

The constraint on $\Psi_{\text{PW}}$ considers the difference between the most and least degraded groups. Therefore, when calculating the gradient of the Lagrangian, only the contribution from said extreme groups appears. This "lack of signal" may make optimization dynamics challenging, illustrated in Fig. 3. The formulation successfully mitigates the disparate impact problem in the context of race prediction for the UTKFace dataset, which features 5 sub-groups. However, when confronted with the CIFAR-100 dataset, encompassing 100 sub-groups, gradient-based approaches to solve Eq. (12) are unable to identify a feasible solution. For both of these scenarios, we observed that the value of the (single) Lagrange multiplier grew continuously, without settling, confirming the previous intuition regarding poor optimization dynamics.

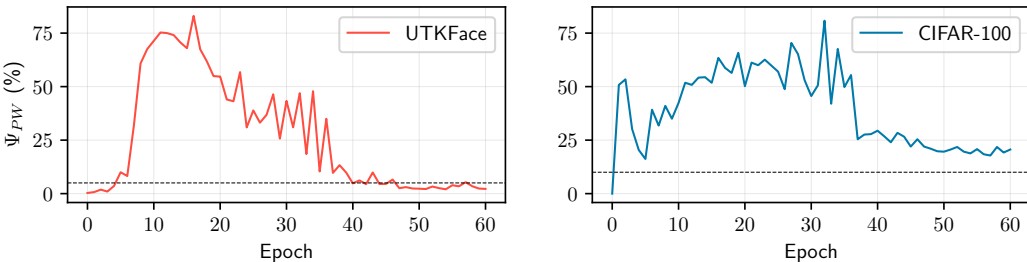

Figure 3: Evolution of disparate impact of pruning ($\Psi_{\text{PW}}$) during training under Eq. (12). **Left:** UTKFace dataset at 92.5% sparsity. **Right:** CIFAR-100 dataset at 95% sparsity. The horizontal dashed lines indicate the tolerance ($\epsilon$) of 5% and 10%, respectively.

A potential approach to alleviate this problem could be to introduce constraints on the pair-wise accuracy gaps:

$$\operatorname*{argmin}_{\boldsymbol{\theta}_s \in \Theta} L(\boldsymbol{\theta}_s | \mathfrak{D}), \qquad \text{s.t.} \ -\epsilon \leq \Delta_g (\boldsymbol{\theta}_s, \boldsymbol{\theta}_d) - \Delta_{g'} (\boldsymbol{\theta}_s, \boldsymbol{\theta}_d) \leq \epsilon, \quad \forall g, g' \in \mathcal{G}. \tag{13}$$

However, this alternative formulation requires quadratically many constraints in the number of protected groups and does not scale to situations where the number of protected groups is large.

**Equalized excess accuracy gaps.** *Equalizing* the per-group excess accuracy gaps to zero gives rise to the following formulation:

$$\operatorname*{argmin}_{\boldsymbol{\theta}_s \in \Theta} L(\boldsymbol{\theta}_s | \mathfrak{D}), \qquad \text{s.t.} \quad \psi_g = \Delta_g (\boldsymbol{\theta}_s, \boldsymbol{\theta}_d) - \Delta (\boldsymbol{\theta}_s, \boldsymbol{\theta}_d) = 0, \quad \forall g \in \mathcal{G}. \tag{14}$$

Compared to CEAG, this formulation (i) does not have an additional tolerance hyper-parameter $\epsilon$, and (ii) prevents groups from having negative EAGs.

However, Eq. (14) can be challenging to solve because it may not have any feasible solutions; equalizing accuracy values may not be possible due to their discrete nature. Moreover, the lack of a tolerance hyper-parameter hurts flexibility as disparity requirements cannot be incorporated into the problem formulation.

**Approximately equal excess accuracy gaps.** A possible way to circumvent the limitations of Eq. (14) is to formulate the constrained problem:

$$\underset{\boldsymbol{\theta}_s \in \Theta}{\text{argmin}}\ L(\boldsymbol{\theta}_s | \mathfrak{D}), \qquad \text{s.t.} \quad |\psi_g| = |\Delta_g(\boldsymbol{\theta}_s, \boldsymbol{\theta}_d) - \Delta(\boldsymbol{\theta}_s, \boldsymbol{\theta}_d)| \leq \epsilon, \quad \forall g \in \mathcal{G}. \tag{15}$$

Feasible solutions of Eq. (15) achieve $\Psi_{\text{PW}} \leq 2\epsilon$ by imposing both an upper and a lower bound on per-group EAGs. Compared to CEAG, this formulation prevents groups from experiencing a large improvement in performance compared to the global accuracy gap. Compared to Eq. (14), it allows for some tolerance at satisfying the equality. Naturally, for reasonable values of $\epsilon$, Eq. (15) has a non-empty set of feasible solutions.

However, since the feasible set of Eq. (15) is small (as prescribed by $\epsilon$), solving it is challenging, especially in the context of stochastic optimization. Mini-batch estimates of the constraints have a high chance of being infeasible due to the small feasible region and noise in the estimation. This leads to updates on the dual variables that are positive most of the time. In turn, this yields dual variables that perpetually increase and never stabilize.

**Two-sided inequality.** Alternatively, a two-sided inequality constrained optimization problem is:

$$\underset{\boldsymbol{\theta}_s \in \Theta}{\text{argmin}}\ L(\boldsymbol{\theta}_s | \mathfrak{D}) \tag{16}$$

$$\text{s.t.} \quad \psi_g = \Delta_g(\boldsymbol{\theta}_s, \boldsymbol{\theta}_d) - \Delta(\boldsymbol{\theta}_s, \boldsymbol{\theta}_d) \leq \epsilon, \quad \forall g \in \mathcal{G} \tag{17}$$

$$-\psi_g = -(\Delta_g(\boldsymbol{\theta}_s, \boldsymbol{\theta}_d) - \Delta(\boldsymbol{\theta}_s, \boldsymbol{\theta}_d)) \leq \epsilon, \quad \forall g \in \mathcal{G}. \tag{18}$$

This problem allows for *individual* dual variables to behave akin to those of CEAG. However, note how the two constraints for each EAG introduce conflicting terms to the gradient of $\boldsymbol{\theta}_s$: the model would aim to increase *or* decrease $\psi_g$ depending on the current values of the dual variables.

**Discussion.** We focus on Eq. (5), and argue that constraining negative EAGs is not crucial for mitigating disparity. A side effect of this choice is allowing for sparse models whose group AGs are arbitrarily below the overall AG. In practice, this may lead to some groups improving their performance while the overall model accuracy decreases. We argue that this behavior is not problematic since it is only likely to manifest for under-represented groups: groups with few samples can deviate in performance from other groups, without significantly influencing overall accuracy.

Eqs. (14) to (16) consider bounds on negative EAGs, but carrying out experiments on them is outside the scope of our work.

## C  REPLAY BUFFERS

---
**Algorithm 2** Update Buffer
---
**Input:** $\text{buf}_g$: Buffer for group $g$, $\hat{\mathbf{y}}$: A batch of model predictions, $\mathbf{y}$: The batch of true targets, $\quad\quad \text{idx}_g$: The sub-group indices of the batch.
1: **function** UPDATEBUFFER($\text{buf}_g, \hat{\mathbf{y}}, \mathbf{y}, \text{idx}_g$)
2: $\quad\quad \text{SampleAcc} \leftarrow (\hat{\mathbf{y}} == \mathbf{y})[\text{idx}_g]$
3: $\quad\quad \text{buf}_g \leftarrow \text{PUSH}(\text{buf}_g, \text{SampleAcc})$ $\quad\quad\quad\quad$ ▷ *Drops old elements to respect capacity $k$*
4: $\quad\quad$ **return** $\text{buf}_g$
5: **end function**
---

Algorithms 2 and 3 contain functions for updating and querying the replay buffers, respectively. These are called by Algorithm 1. Note that we wait until a buffer has been filled before considering its contents for computing the $\psi_g$ terms. Before a buffer is filled, its corresponding $\psi_g$ is 0.

For all groups, we consider the same buffer memory size $k$. Thus, the effective dataset used when computing EAGs of the sparse model is balanced across groups: it has $k$ samples per group. However, when the original dataset is not balanced, this design implies that over-represented classes refresh their buffers faster as opposed to under-represented classes.

---

**Algorithm 3** Query Buffers

---

**Input:** $\texttt{buf}_g, \forall g \in \mathcal{G}$: All replay buffers, $k$: Memory size for the replay buffers, $A^g_{\text{dense}}$: Accuracy of the dense model on each group $g$, $A_{\text{dense}}$: Aggregate accuracy of the dense model.

1: **function** QUERYBUFFERS($\{\texttt{buf}_g\}^{\mathcal{G}}_{g=1}, k, \{A^g_{\text{dense}}\}^{\mathcal{G}}_{g=1}, A_{\text{dense}}$)
2:      $\mathcal{I} \leftarrow \{\}$             *▷ Indices of full buffers*
3:      **for** $g \in \mathcal{G}$ **do**
4:          **if** LEN($\texttt{buf}_g$) $== k$ **then**
5:              $\mathcal{I} \leftarrow \mathcal{I} \cup \{g\}$
6:              $\texttt{SampleAcc}_g \leftarrow$ QUERY($\texttt{buf}_g, k$)          *▷ Query all elements of each buffer*
7:              $A^g_{\text{sparse}} \leftarrow$ AVERAGE($\texttt{SampleAcc}_g$)
8:          **end if**
9:      **end for**
10:     $A_{\text{sparse}} \leftarrow$ AVERAGE($\{A^g_{\text{sparse}}\}_{g \in \mathcal{I}}$)          *▷ Compute aggregate accuracy from full buffers*
11:     **for** $g \in \mathcal{G}$ **do**
12:         **if** $g \in \mathcal{I}$ **then**
13:             $\psi_g \leftarrow (A^g_{\text{sparse}} - A^g_{\text{dense}}) - (A_{\text{sparse}} - A_{\text{dense}})$
14:         **else**
15:             $\psi_g \leftarrow 0$          *▷ Ignore non-full buffers in $\psi_g$*
16:         **end if**
17:     **end for**
18:     **return** $\{\psi_g\}^{\mathcal{G}}_{g=1}$
19: **end function**

---

### C.1 TRAINING DYNAMICS WITH REPLAY BUFFERS

In Fig. 4, we present the behavior of a select multiplier in a CIFAR-100 experiment with 90% sparsity. We depict two training stages: on the left, the multiplier consistently maintains a non-zero value, while on the right, it is frequently around zero. Multipliers are initialized at zero, and are expected to increase during the first stages of training when constraints may not be satisfied. Moreover, they are expected to eventually stabilize at a value (possibly zero) if their corresponding constraint is inactive at the solution.

On the left plot, we observe a smooth curve for the dual variable corresponding to the run with replay buffers. In contrast, the dual variable for the run without buffers is more noisy. On the right plot, the multiplier associated with the run without buffers becomes active more frequently than the multiplier of the run with buffers. Given the small magnitude of these multipliers (up to 0.003), the constraint may actually be feasible in this region and so the desirable behavior is to keep multipliers at 0.

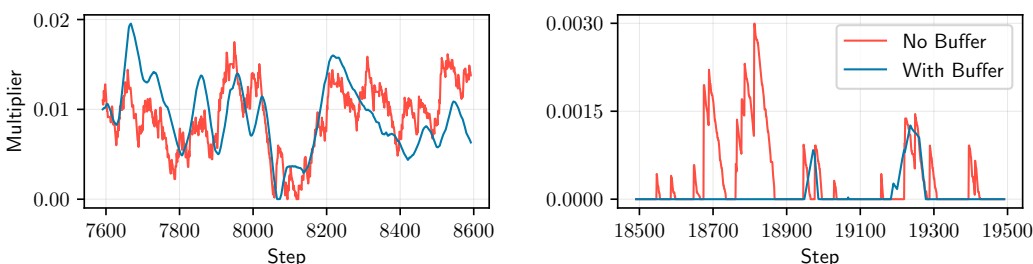

Figure 4: Effects of replay buffers on the multiplier dynamics on CIFAR-100 under 90% sparsity. As expected, the multiplier exhibits notably smoother dynamics when using replay buffers.

### C.2 REPLAY BUFFER SIZE ABLATION

Table 4 showcases the effect of the choice of buffer size in terms of accuracy and disparate impact. We observe that having a buffer is beneficial in terms of the train and test $\max_g \psi_g$, while yielding models with similar accuracy to those obtained without replay buffers.

Table 4: Effects of the memory size of replay buffers on a CIFAR-100 task at 95% sparsity. Not using a buffer yields poor results in terms of $\max_g \psi_g$. For experiments with buffers, different choices of memory sizes yield comparable results. We consider a tolerance of $\epsilon = 5\%$.

| Buffer Size ($k$) | Train | | Test | |
|---|---|---|---|---|
| | Accuracy | $\max_g \psi_g$ | Accuracy | $\max_g \psi_g$ |
| No Buffer | $95.8 \pm 0.15$ | $5.8 \pm 0.53$ | $62.5 \pm 0.41$ | $17.1 \pm 3.59$ |
| 20 | $95.7 \pm 0.10$ | $5.4 \pm 0.69$ | $62.6 \pm 0.30$ | $16.0 \pm 2.82$ |
| 40 | $95.6 \pm 0.12$ | $5.7 \pm 0.49$ | $62.7 \pm 0.28$ | $14.8 \pm 1.52$ |
| 60 | $95.6 \pm 0.16$ | $5.5 \pm 0.63$ | $62.7 \pm 0.26$ | $14.5 \pm 1.92$ |
| 80 | $95.6 \pm 0.17$ | $5.5 \pm 0.42$ | $62.8 \pm 0.44$ | $16.4 \pm 4.59$ |

We observe that changing the buffer size has a small impact in terms of accuracy. In terms of $\max_g \psi_g$, the smallest (20) and largest (80) choices of buffer size result in more significant overfitting compared to 40 and 60. Moreover, the maximum EAG in test shows high variance in these cases. Table 4 motivates our choice of $k = 40$ for most experiments.

# D  EXPERIMENTAL DETAILS

Our implementations are in PyTorch 1.13.0 (Paszke et al., 2019), with the Cooper library for Lagrangian-based constrained optimization (Gallego-Posada & Ramirez, 2022).

**Pipeline.** As illustrated in Fig. 1, our pipeline consists of 3 stages: (i) obtaining a dense pre-trained model, (ii) pruning said model using gradual magnitude pruning, and (iii) fine-tuning the sparse model using either empirical risk minimization, the equalized loss formulation of Tran et al. (2022), or our approach.

**Dense models.** Except for tasks involving the UTKFace dataset, we use publicly accessible pre-trained dense models. Appendix D.4 provides references to the pre-trained models we use throughout this work.

**Pruning.** We perform unstructured, layer-wise, gradual magnitude pruning (Zhu & Gupta, 2017) with a cubic sparsity schedule (see Appendix D.5). We sparsify the weights of the model, but not the biases. We also do not sparsify the input and output layers of the model, as recommended by Gale et al. (2019). See more details in Appendix D.3.

**Tolerance level $\epsilon$.** We choose the tolerance level for each experiment by running NFT, measuring its corresponding $\max_g \psi_g$ and choosing a value of $\epsilon$ below this level. This protocol is ran independently for every task and every sparsity level. Finally, note that since EL imposes an equality constraint, there is no tolerance hyper-parameter to be chosen.

## D.1  TASKS AND PROTECTED ATTRIBUTES

We carry out experiments on the UTKFace (Zhang et al., 2017), FairFace (Kärkkäinen & Joo, 2021), and CIFAR-100 (Krizhevsky, 2009) datasets; these respectively employ MobileNet-V2 (Sandler et al., 2018), ResNet-34 (He et al., 2016), and CifarResNet-56 models (Chen, 2021), using different sparsity levels. These details are summarized in Table 5.

We highlight the data transformations and the batch size we employ for each dataset in Table 6.

## D.2  MITIGATION SCHEMES

This section describes the approaches considered throughout this work for fine-tuning the sparse model, with or without a scheme to mitigate the disparate impact of pruning. We fine-tune sparse models on UTKFace and CIFAR for 45 epochs, and for 32 epochs on FairFace.

**Naive Fine Tuning (NFT).** The sparse model is fine-tuned on the training set using ERM.

Table 5: Tasks considered throughout this work.

| Dataset | Model | Predicted Attribute | Group Attribute | Sparsity |
|---------|-------|---------------------|-----------------|----------|
| UTKFace | MobileNet-V2 | Race
Gender
Race | Race
Race
Race $\cap$ Gender | $85, 90, 92.5$
$85, 90, 92.5$
$85, 90, 92.5$ |
| FairFace | ResNet-34 | Race
Gender
Race | Race
Race
Race $\cap$ Gender | $99$
$99$
$99$ |
| CIFAR-100 | CifarResNet-56 | Class | Class | $90, 92.5, 95$ |

Table 6: Transformations and batch sizes considered for each dataset.

| Dataset | Transformations | | Batch Size |
|---------|-----------------|------|------------|
| | Train | Test | |
| UTKFace | `RandomHorizontalFlip(0.5)` | – | `128` |
| FairFace | `Resize(224,224)`
`RandomHorizontalFlip(0.5)` | `Resize(224,224)` | `256` |
| CIFAR | – | – | `128` |

**Naive Fine Tuning with Early Stopping (NFT+ES).** Obtained by selecting the best iterate of NFT in terms of *test* accuracy. We analyze this approach since early stopping is a popular technique in deep learning practice and, as evidenced by our experiments, often results in higher disparity (compared to the last iterate in NFT).

**EL.** Our implementation of the equalized loss method proposed by Tran et al. (2022). More details of this formulation can be found in Appendix B.1.

**EL+RB.** Enhanced version of EL employing replay buffers (§4.2) for updating the dual variables. The replay buffers store the per-sample losses observed at the mini-batch level across groups.

**CEAG.** Our constrained excess accuracy gap approach (see §4.3), which uses replay buffers by default.

### D.3 MODEL ARCHITECTURES

We employ MobileNet-V2 (Sandler et al., 2018), ResNet-34, and CifarResNet-56 models (He et al., 2016). ResNet-34 models are composed of bottleneck residual blocks He et al. (2016), while CifarResNet-56 models use basic residual blocks (Chen, 2021).

Following Evci et al. (2020), across all models, we do not sparsify the biases due to their low footprint towards the total number of parameters. We also do not sparsify the first and last layers of the model as recommended by Gale et al. (2019).

Table 7 specifies the number of parameters of all considered architectures. We also provide the number of parameters remaining post-pruning across the considered sparsities for the reader's convenience.

### D.4 PRE-TRAINED MODELS

Reusing and fine-tuning of pre-trained deep learning models is a common practice. For example, a typical application pipeline might involve (i) obtaining a pre-trained model, (ii) fine-tuning it on an application-specific task, and (iii) pruning it before deployment.

Therefore, we concentrate on studying the behavior of mitigation techniques when applied to openly available pre-trained models.

Table 7: Statistics on the total number of parameters and active parameters at different sparsity levels for our employed architectures. [†]Sparsifiable parameters indicate the number of parameters that *may* be removed during pruning (thus, excluding non-prunable parameters such as biases). [‡]Parameter counts reported for MobileNet-V2 and ResNet-34 models are for race prediction tasks on UTKFace and FairFace, respectively.

| Architecture | Total | Sparsifiable | Active parameters at sparsity: | | | | |
|---|---|---|---|---|---|---|---|
| | Params | Params[†] | 85% | 90% | 92.5% | 95% | 99% |
| MobileNet–V2[‡] | 2,230,277 | 2,222,944 | 366,946 | 255,250 | 198,709 | – | – |
| ResNet-34[‡] | 21,288,263 | 21,275,136 | – | – | – | – | 241,751 |
| CifarResNet-56 | 861,620 | 854,656 | – | 96,179 | 74,889 | 53,621 | – |

- ResNet-34 models for FairFace use the weights provided by Kärkkäinen & Joo (2021).
- CifarResNet-56 models for CIFAR-100 use the weights provided by Chen (2021).
- We were unable to find publicly available pre-trained MobileNet-V2 models for the UTKFace dataset. Thus, we train these from scratch (see details below). *As part of our reproducibility efforts, we are making our pre-trained UTKFace MobileNet-V2 models openly available.*

For training UTKFace models, we use SGD with an initial learning rate of $0.01$, decayed by a factor of $0.1$ at training milestones of $60\%$, $80\%$, and $90\%$ of total training epochs. We use a momentum coefficient of $0.9$, and train for a total of $50$ epochs. These hyper-parameters are used both for race and gender prediction tasks.

The group-wise performance for all the dense models is reported in Tables 17, 18, 20, 21, 23, 24, 26, 27, 29, 30, 32 and 33.

### D.5 GRADUAL MAGNITUDE PRUNING

As mentioned in §5.1, our experiments perform Gradual Magnitude Pruning (GMP), where a fraction of the smallest weights (in magnitude) is pruned on every epoch. Zhu & Gupta (2017) consider the following cubic schedule prescribing the proportion of parameters to prune at every epoch:

$$s_t = s_f + (s_i - s_f) \left( 1 - \frac{t - t_0}{(T_{\text{end}} - t_0)\,\Delta t} \right)^3 \quad t \in \{t_0, t_0 + \Delta t, ..., t_0 + (T_{\text{end}} - t_0)\Delta t\}, \quad (19)$$

where $t_0$ is the initial training step, $\Delta t$ is the pruning frequency (in epochs), $T_{\text{end}}$ is final epoch of pruning, and $s_i$ and $s_f$ denote the initial and final sparsities, respectively.

Our experiments carry out GMP since epoch $t_0 = 0$, throughout $T_{\text{end}} = 15 - 1 = 14$ epochs, and perform pruning once every epoch ($\Delta t = 1$).

### D.6 PRIMAL OPTIMIZATION HYPER-PARAMETERS

We make use of SGD with momentum as the primal optimizer for all of our experiments. In our initial ablation experiments on the choice of primal optimizer, we found that employing SGD with momentum outperformed or matched the performance of Adam (Kingma & Ba, 2014).

For UTKFace and CIFAR-100 datasets, we employ a primal step size of $1 \cdot 10^{-2}$ along with a momentum of 0.9 (Polyak), and apply weight decay at the rate of $1 \cdot 10^{-4}$.

For FairFace, we employ Nesterov momentum with a step-size of $1 \cdot 10^{-3}$ and apply a weight decay of $1 \cdot 10^{-2}$. Specifically, for race prediction tasks, we utilize a momentum of 0.95, while for gender prediction tasks, a momentum of 0.99 is employed.

Additionally, we use PyTorch's `MultiStepLR` as the learning rate scheduler, with decay $\gamma = 0.1$ across all experiments. For UTKFace and CIFAR-100, we set scheduler milestones at 60%, 80% and 90% of the total training epochs (including the execution of GMP). For instance, for a task that has 60 epochs where it employs GMP on 15 epochs, the above milestones would activate at epoch 36, epoch 48 and epoch 54. For race prediction on FairFace we use a single milestone at 90%, while gender prediction on FairFace uses a constant learning rate of $1 \cdot 10^{-2}$.

## D.7 Dual Optimization Hyper-parameters

We employ stochastic gradient *ascent* on the dual parameters (corresponding to the Lagrange multipliers) in all experiments. The choices of dual learning rate are presented in Tables 8 to 13.

We fix $k = 40$ as the memory size for the replay buffer. Preliminary ablations on the choice of $k \in [20, 80]$ showed low sensitivity to the specific value of this hyper-parameter (See Appendix C.2).

Note that the order of magnitude for the dual step-size choices is relatively consistent across datasets, tasks, sparsity levels and disparity tolerances. This highlights the ease of tuning exhibited by this hyper-parameter.

### D.7.1 UTKFACE

Table 8: Tolerance and dual step-size for CEAG on UTKFace tasks.

| Target Attribute | Group Attribute | Sparsity | Dual Step-Size ($\eta_\lambda$) | Tolerance $\epsilon$ (%) |
|---|---|---|---|---|
| Gender | Race | 85 | $2 \cdot 10^{-3}$ | 0.5 |
| | | 90 | $1 \cdot 10^{-4}$ | 0.5 |
| | | 92.5 | $3 \cdot 10^{-3}$ | 0.5 |
| Race | Race | 85 | $1 \cdot 10^{-4}$ | 0.25 |
| | | 90 | $2 \cdot 10^{-3}$ | 1 |
| | | 92.5 | $2 \cdot 10^{-3}$ | 1 |
| Race | Race $\cap$ Gender | 85 | $1 \cdot 10^{-5}$ | 0.5 |
| | | 90 | $1 \cdot 10^{-3}$ | 3 |
| | | 92.5 | $1 \cdot 10^{-3}$ | 3 |

Table 9: Dual step-size for EL+RB on UTKFace tasks.

| Target Attribute | Group Attribute | Sparsity | Dual Step-Size ($\eta_\lambda$) |
|---|---|---|---|
| Gender | Race | 85 | $1 \cdot 10^{-4}$ |
| | | 90 | $1 \cdot 10^{-5}$ |
| | | 92.5 | $1 \cdot 10^{-5}$ |
| Race | Race | 85 | $1 \cdot 10^{-5}$ |
| | | 90 | $1 \cdot 10^{-5}$ |
| | | 92.5 | $1 \cdot 10^{-5}$ |
| Race | Race $\cap$ Gender | 85 | $1 \cdot 10^{-5}$ |
| | | 90 | $1 \cdot 10^{-5}$ |
| | | 92.5 | $1 \cdot 10^{-5}$ |

### D.7.2 FAIRFACE

Table 10: Tolerance and dual step-size for CEAG on FairFace tasks at 99% sparsity.

| Target Attribute | Group Attribute | Dual Step-Size ($\eta_\lambda$) | Tolerance $\epsilon$ (%) |
|---|---|---|---|
| Gender | Race | $1 \cdot 10^{-5}$ | 1 |
| Race | Race | $1 \cdot 10^{-4}$ | 2 |
| Race | Race $\cap$ Gender | $1 \cdot 10^{-5}$ | 0.25 |

Table 11: Dual step-size for EL+RB on FairFace tasks.

| Target Attribute | Group Attribute | Dual Step-Size ($\eta_\lambda$) |
|---|---|---|
| Gender | Race | $1 \cdot 10^{-5}$ |
| Race | Race | $1 \cdot 10^{-5}$ |
| Race | Race $\cap$ Gender | $1 \cdot 10^{-5}$ |

### D.7.3 CIFAR

Table 12: Tolerance and dual step-size for CEAG on CIFAR tasks.

| Target Attribute | Group Attribute | Sparsity | Dual Step-Size ($\eta_\lambda$) | Tolerance $\epsilon$ (%) |
|---|---|---|---|---|
| Class | Class | 90 | $2 \cdot 10^{-3}$ | 1 |
| | | 92.5 | $2 \cdot 10^{-3}$ | 2 |
| | | 95 | $1 \cdot 10^{-3}$ | 5 |

Table 13: Dual step-size for EL+RB on CIFAR tasks.

| Target Attribute | Group Attribute | Sparsity | Dual Step-Size ($\eta_\lambda$) |
|---|---|---|---|
| Class | Class | 90 | $1 \cdot 10^{-5}$ |
| | | 92.5 | $1 \cdot 10^{-5}$ |
| | | 95 | $1 \cdot 10^{-5}$ |

## E  ADDITIONAL EXPERIMENTS

### E.1  COMPUTATIONAL OVERHEAD

Table 14 presents the wall-clock time of an experiment for different mitigation approaches on CIFAR-100 at 95% sparsity. Note that the reported time includes the 15 epochs of gradual magnitude pruning of the dense model, as well as the 45 epochs of fine-tuning.

Table 14: Runtime of different mitigation approaches on CIFAR-100 at 95% sparsity. All runs are run on NVIDIA `A100-SXM4-80GB` GPUs. Runtimes are average across 5 runs for each mitigation method.

| Method | Min | | Median | | Max | |
|---|---|---|---|---|---|---|
| | Wall-clock Time | Overhead wrt NFT | Wall-clock Time | Overhead wrt NFT | Wall-clock Time | Overhead wrt NFT |
| NFT | 1h 0m 04s | $1\times$ | 1h 3m 13s | $1\times$ | 1h 12m 19s | $1\times$ |
| EL | 1h 2m 37s | $1.031\times$ | 1h 4m 34s | $1.021\times$ | 1h 15m 50s | $1.049\times$ |
| EL+RB | 1h 4m 15s | $1.058\times$ | 1h 7m 10s | $1.062\times$ | 1h 17m 35s | $1.073\times$ |
| CEAG (No RB) | 1h 2m 08s | $1.023\times$ | 1h 3m 08s | $0.998\times$ | 1h 8m 35s | $0.948\times$ |
| CEAG | 1h 1m 58s | $1.020\times$ | 1h 4m 28s | $1.020\times$ | 1h 6m 27s | $0.919\times$ |

We observe a negligible increase in training time for constrained approaches that use replay buffers relative to NFT. For approaches that do not use replay buffers, the runtime is essentially the same as NFT. This overhead is especially insignificant considering that the CIFAR-100 problem involves 100 constraints.

### E.2 COMPARISON TO FAIRGRAPE (LIN ET AL., 2022)

**Computational cost**: As a benchmarking exercise, we re-ran the code provided by Lin et al. (2022) for UTKFace Race prediction and Race as protected group[5]. The method took more than 90 hours of compute time on an NVIDIA `A100-SXM4-80GB` GPU. Given how prohibitively expensive this is, we refrained from running experiments with FairGRAPE. Furthermore, we expect the runtime to increase for tasks with larger numbers of protected groups.

**UTKFace**: Lin et al. (2022) apply their method on UTKFace, but remove race group *Others* from the dataset. The authors state that this was done as *Others* is an ambiguous class. Since we consider the complete dataset in our experiments, we can not compare directly to the numbers reported by Lin et al. (2022).

Although we could apply CEAG to the UTKFace dataset without race group *Others*, we choose not to since we observe that this group is generally the most disproportionately affected by pruning. Table 15 shows that NFT can achieve models with low disparity on UTKFace without *Others*, but presents significantly worse accuracy and higher disparity on experiments with *Others*.

Table 15: NFT results on UTKFace race prediction with race as group attribute, with and without race group *Others*.

| Setup | Train | | | Test | | |
|---|---|---|---|---|---|---|
| | **Accuracy** | $\Psi_{\text{PW}}$ | $\max_g \psi_g$ | **Accuracy** | $\Psi_{\text{PW}}$ | $\max_g \psi_g$ |
| UTKFace (without *Others*) | $99.5 \pm 0.0$ | $2.0 \pm 0.16$ | $0.5 \pm 0.00$ | $86.7 \pm 0.55$ | $10.9 \pm 1.68$ | $7.4 \pm 0.14$ |
| UTKFace | $98.2 \pm 0.0$ | $9.9 \pm 0.82$ | $8.7 \pm 0.82$ | $79.5 \pm 0.46$ | $6.1 \pm 1.60$ | $2.2 \pm 0.68$ |

### E.3 SENSITIVITY ANALYSIS

Table 16 presents the sensitivity of our approach to the tolerance hyperparameter $\epsilon$ on a UTKFace race prediction task with race as group attribute.

Table 16: Race prediction task for UTKFace with race as group attribute, at 92.5% sparsity. All experiments use a dual step size of $2 \cdot 10^{-3}$. Results are aggregated across 5 seeds.

| Tolerance | Accuracy | $\max_g \psi_g$ |
|---|---|---|
| 1.5 | $93.6 \pm 0.1$ | $0.9 \pm 0.61$ |
| 1.0 | $93.4 \pm 0.3$ | $1.1 \pm 0.36$ |
| 0.5 | $93.2 \pm 0.2$ | $1.2 \pm 0.44$ |
| 0 | $93.2 \pm 0.2$ | $0.9 \pm 0.24$ |

We observe small improvements in performance for experiments with large tolerance values. For low tolerance values, we observe that feasibility is not attained. Moreover, the resulting $\max_g \psi_g$ values are similar across the considered tolerances. These observations are indicative of robustness to the choice of tolerance.

## F COMPREHENSIVE EXPERIMENTAL RESULTS

This section contains the results corresponding to all experiments mentioned in Table 5 across all datasets, sparsities, and tasks considered in our work. As mentioned earlier, all metrics reported in our tables and plots follow the pattern `avg ± std`. Unless mentioned otherwise, all our experimental metrics are aggregated across 5 seeds.

Some of the tables displayed below are extensions of tables presented in the main paper. Such tables have been clearly identified in the captions. For the tables reporting group accuracies, we report the numbers for both the dense model as well as all the sparse models.

---

[5]The FairGRAPE implementation can be found here: `https://github.com/Bernardo1998/FairGRAPE`

## F.1 UTKFACE

We use MobileNet-V2 Sandler et al. (2018), similar to Lin et al. (2022).

### F.1.1 GENDER

Table 17: Groupwise train accuracy for gender prediction in UTKFace with race as protected attribute, across sparsities.

| Sparsity | Method | White | Black | Asian | Indian | Others |
|---|---|---|---|---|---|---|
| 85 | Dense | 100.0 | 99.9 | 99.9 | 99.8 | 99.3 |
| | NFT | $99.9 \pm 0.02$ | $99.8 \pm 0.04$ | $99.9 \pm 0.06$ | $99.8 \pm 0.02$ | $99.3 \pm 0.1$ |
| | EL + RB | $99.9 \pm 0.01$ | $99.8 \pm 0.03$ | $99.9 \pm 0.03$ | $99.8 \pm 0.04$ | $99.4 \pm 0.13$ |
| | CEAG | $99.9 \pm 0.03$ | $99.8 \pm 0.06$ | $99.9 \pm 0.05$ | $99.8 \pm 0.03$ | $99.3 \pm 0.12$ |
| 90 | Dense | 100.0 | 99.9 | 99.9 | 99.8 | 99.3 |
| | NFT | $99.8 \pm 0.03$ | $99.8 \pm 0.04$ | $99.7 \pm 0.09$ | $99.6 \pm 0.1$ | $98.9 \pm 0.17$ |
| | EL + RB | $99.8 \pm 0.05$ | $99.8 \pm 0.05$ | $99.7 \pm 0.09$ | $99.6 \pm 0.08$ | $99.0 \pm 0.17$ |
| | CEAG | $99.8 \pm 0.02$ | $99.8 \pm 0.02$ | $99.7 \pm 0.12$ | $99.7 \pm 0.04$ | $99.1 \pm 0.1$ |
| 92.5 | Dense | 100.0 | 99.9 | 99.9 | 99.8 | 99.3 |
| | NFT | $99.4 \pm 0.07$ | $99.5 \pm 0.08$ | $98.8 \pm 0.22$ | $99.1 \pm 0.11$ | $98.0 \pm 0.23$ |
| | EL + RB | $99.5 \pm 0.05$ | $99.6 \pm 0.09$ | $98.6 \pm 0.23$ | $99.1 \pm 0.1$ | $98.1 \pm 0.45$ |
| | CEAG | $99.1 \pm 0.22$ | $99.3 \pm 0.28$ | $99.6 \pm 0.12$ | $98.7 \pm 0.11$ | $98.5 \pm 0.29$ |

Table 18: Groupwise test accuracy for gender prediction in UTKFace with race as protected attribute, across sparsities.

| Sparsity | Method | White | Black | Asian | Indian | Others |
|---|---|---|---|---|---|---|
| 85 | Dense | 94.2 | 95.0 | 89.5 | 93.2 | 89.5 |
| | NFT | $92.0 \pm 0.42$ | $94.0 \pm 0.82$ | $87.0 \pm 0.62$ | $92.5 \pm 0.37$ | $88.6 \pm 0.83$ |
| | EL + RB | $92.1 \pm 0.45$ | $94.2 \pm 0.52$ | $87.2 \pm 0.72$ | $92.1 \pm 0.41$ | $88.4 \pm 0.4$ |
| | CEAG | $91.9 \pm 0.3$ | $93.8 \pm 0.41$ | $86.7 \pm 0.75$ | $92.1 \pm 0.55$ | $88.1 \pm 1.03$ |
| 90 | Dense | 94.2 | 95.0 | 89.5 | 93.2 | 89.5 |
| | NFT | $91.1 \pm 0.67$ | $92.6 \pm 0.57$ | $85.9 \pm 0.95$ | $91.6 \pm 0.67$ | $87.1 \pm 0.85$ |
| | EL + RB | $91.0 \pm 0.1$ | $93.0 \pm 0.38$ | $86.4 \pm 0.64$ | $91.4 \pm 0.45$ | $87.6 \pm 0.83$ |
| | CEAG | $91.0 \pm 0.56$ | $93.2 \pm 0.64$ | $85.5 \pm 0.73$ | $91.8 \pm 0.64$ | $86.3 \pm 1.03$ |
| 92.5 | Dense | 94.2 | 95.0 | 89.5 | 93.2 | 89.5 |
| | NFT | $90.5 \pm 0.67$ | $92.6 \pm 0.62$ | $85.1 \pm 0.83$ | $91.0 \pm 0.89$ | $87.3 \pm 1.05$ |
| | EL + RB | $90.2 \pm 0.54$ | $93.2 \pm 0.66$ | $85.0 \pm 1.57$ | $90.6 \pm 0.4$ | $86.6 \pm 1.35$ |
| | CEAG | $90.6 \pm 1.0$ | $92.7 \pm 0.2$ | $85.8 \pm 0.93$ | $90.5 \pm 0.76$ | $87.1 \pm 0.91$ |

Table 19: Gender prediction on UTKFace with race as group attribute, across sparsities. **CEAG consistently achieves a $\max_g \psi_g$ within the threshold, across sparsities**.

| Sparsity | Method | Train | | | | Test | | |
|---|---|---|---|---|---|---|---|---|
| | | Accuracy | $\Psi_{PW}$ | $\max_g \psi_g$ | Tol ($\epsilon$) | Accuracy | $\Psi_{PW}$ | $\max_g \psi_g$ |
| 85 | NFT | $99.8 \pm 0.01$ | $0.3 \pm 0.15$ | $0.0 \pm 0.02$ | – | $91.5 \pm 0.25$ | $2.2 \pm 0.87$ | $0.9 \pm 0.43$ |
| | NFT + ES | $98.0 \pm 1.72$ | $1.7 \pm 1.19$ | $1.3 \pm 0.91$ | – | $91.8 \pm 0.28$ | $2.1 \pm 0.69$ | $0.8 \pm 0.23$ |
| | EL + RB | $99.9 \pm 0.02$ | $0.3 \pm 0.18$ | $0.0 \pm 0.01$ | – | $91.5 \pm 0.29$ | $1.9 \pm 0.71$ | $0.9 \pm 0.41$ |
| | CEAG | $99.8 \pm 0.01$ | $0.3 \pm 0.16$ | $0.1 \pm 0.05$ | $\leq 0.5\%$ ✓ | $91.3 \pm 0.3$ | $2.1 \pm 0.78$ | $0.9 \pm 0.51$ |
| 90 | NFT | $99.7 \pm 0.04$ | $0.4 \pm 0.26$ | $0.3 \pm 0.2$ | – | $90.5 \pm 0.2$ | $2.7 \pm 1.03$ | $1.3 \pm 0.65$ |
| | NFT + ES | $97.4 \pm 1.59$ | $2.5 \pm 1.42$ | $1.8 \pm 1.15$ | – | $91.0 \pm 0.15$ | $1.8 \pm 0.9$ | $1.0 \pm 0.65$ |
| | EL + RB | $99.7 \pm 0.05$ | $0.3 \pm 0.16$ | $0.2 \pm 0.15$ | – | $90.6 \pm 0.17$ | $2.0 \pm 0.52$ | $0.8 \pm 0.27$ |
| | CEAG | $99.7 \pm 0.03$ | $0.2 \pm 0.1$ | $0.2 \pm 0.05$ | $\leq 0.5\%$ ✓ | $90.4 \pm 0.37$ | $3.0 \pm 0.82$ | $1.4 \pm 0.51$ |
| 92.5 | NFT | $99.2 \pm 0.08$ | $1.0 \pm 0.17$ | $0.7 \pm 0.19$ | – | $90.0 \pm 0.44$ | $3.1 \pm 0.71$ | $1.4 \pm 0.57$ |
| | NFT + ES | $96.2 \pm 1.82$ | $3.0 \pm 0.98$ | $2.2 \pm 0.8$ | – | $90.4 \pm 0.37$ | $2.8 \pm 0.55$ | $1.2 \pm 0.52$ |
| | EL + RB | $99.2 \pm 0.07$ | $1.3 \pm 0.34$ | $0.9 \pm 0.3$ | – | $89.8 \pm 0.29$ | $3.1 \pm 1.64$ | $1.5 \pm 0.98$ |
| | CEAG | $99.1 \pm 0.14$ | $0.8 \pm 0.24$ | $0.4 \pm 0.09$ | $\leq 0.5\%$ ✓ | $90.1 \pm 0.52$ | $2.2 \pm 0.75$ | $1.0 \pm 0.2$ |

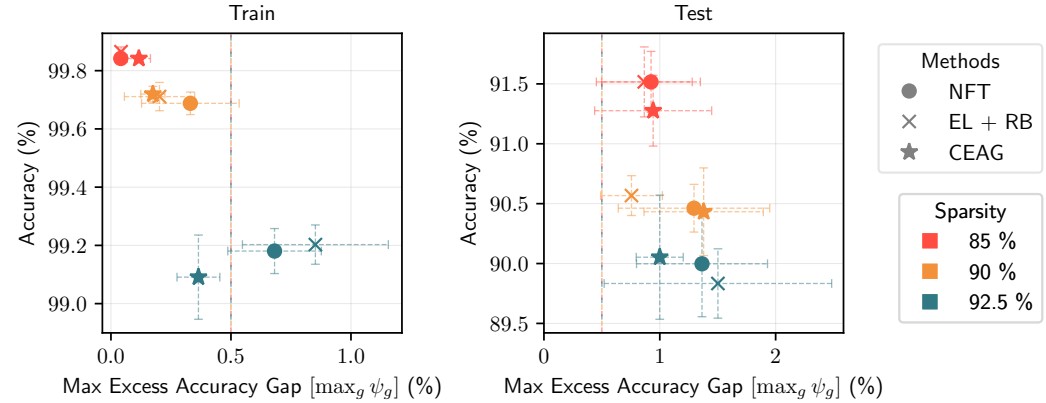

Figure 5: UTKFace gender prediction with race as protected attribute.

F.1.2  RACE

Table 20: Groupwise train accuracy for race prediction in UTKFace with race as protected attribute, across sparsities.

| Sparsity | Method | White | Black | Asian | Indian | Others |
|---|---|---|---|---|---|---|
| 85 | Dense | 99.8 | 99.7 | 99.9 | 99.8 | 99.5 |
| | NFT | $99.7 \pm 0.08$ | $99.6 \pm 0.11$ | $99.8 \pm 0.06$ | $99.7 \pm 0.06$ | $99.0 \pm 0.13$ |
| | EL + RB | $99.6 \pm 0.11$ | $99.6 \pm 0.18$ | $99.8 \pm 0.1$ | $99.8 \pm 0.14$ | $99.5 \pm 0.1$ |
| | CEAG | $99.6 \pm 0.13$ | $99.5 \pm 0.17$ | $99.7 \pm 0.09$ | $99.7 \pm 0.19$ | $99.8 \pm 0.09$ |
| 90 | Dense | 99.8 | 99.7 | 99.9 | 99.8 | 99.5 |
| | NFT | $98.6 \pm 0.26$ | $99.2 \pm 0.24$ | $99.3 \pm 0.04$ | $98.9 \pm 0.2$ | $89.0 \pm 0.79$ |
| | EL + RB | $98.4 \pm 0.19$ | $98.8 \pm 0.18$ | $99.2 \pm 0.22$ | $98.7 \pm 0.3$ | $94.3 \pm 0.61$ |
| | CEAG | $95.7 \pm 0.35$ | $95.5 \pm 0.35$ | $96.0 \pm 0.4$ | $97.1 \pm 0.52$ | $98.4 \pm 0.42$ |
| 92.5 | Dense | 99.8 | 99.7 | 99.9 | 99.8 | 99.5 |
| | NFT | $96.8 \pm 0.15$ | $98.0 \pm 0.23$ | $98.2 \pm 0.44$ | $96.3 \pm 0.47$ | $69.4 \pm 3.72$ |
| | EL + RB | $95.9 \pm 0.37$ | $97.2 \pm 0.23$ | $97.6 \pm 0.55$ | $95.9 \pm 0.56$ | $82.5 \pm 1.36$ |
| | CEAG | $93.2 \pm 0.28$ | $92.9 \pm 0.74$ | $92.8 \pm 0.99$ | $93.9 \pm 0.79$ | $95.4 \pm 0.35$ |

Table 21: Groupwise test accuracy for race prediction in UTKFace with race as protected attribute, across sparsities.

| Sparsity | Method | White | Black | Asian | Indian | Others |
|---|---|---|---|---|---|---|
| 85 | Dense | 90.6 | 87.9 | 88.5 | 80.7 | 29.2 |
| | NFT | $87.3 \pm 0.24$ | $84.4 \pm 1.32$ | $86.7 \pm 0.82$ | $74.9 \pm 1.01$ | $31.0 \pm 1.55$ |
| | EL + RB | $87.2 \pm 0.48$ | $84.1 \pm 0.74$ | $86.1 \pm 1.33$ | $75.2 \pm 1.53$ | $32.8 \pm 1.47$ |
| | CEAG | $86.5 \pm 0.31$ | $84.3 \pm 1.31$ | $85.7 \pm 1.25$ | $75.3 \pm 1.26$ | $32.1 \pm 2.36$ |
| 90 | Dense | 90.6 | 87.9 | 88.5 | 80.7 | 29.2 |
| | NFT | $86.5 \pm 0.52$ | $83.3 \pm 1.28$ | $84.2 \pm 2.46$ | $74.6 \pm 0.44$ | $28.8 \pm 1.95$ |
| | EL + RB | $86.2 \pm 0.94$ | $83.7 \pm 0.91$ | $83.6 \pm 1.15$ | $75.3 \pm 1.08$ | $31.4 \pm 1.33$ |
| | CEAG | $86.6 \pm 0.83$ | $84.2 \pm 1.1$ | $85.6 \pm 1.12$ | $77.7 \pm 1.07$ | $29.1 \pm 2.73$ |
| 92.5 | Dense | 90.6 | 87.9 | 88.5 | 80.7 | 29.2 |
| | NFT | $86.3 \pm 0.78$ | $83.8 \pm 0.58$ | $84.6 \pm 0.8$ | $73.8 \pm 1.04$ | $28.5 \pm 2.52$ |
| | EL + RB | $85.1 \pm 1.0$ | $82.8 \pm 1.37$ | $83.5 \pm 1.49$ | $73.4 \pm 1.09$ | $30.9 \pm 2.49$ |
| | CEAG | $85.5 \pm 0.6$ | $83.0 \pm 0.65$ | $84.6 \pm 1.66$ | $76.3 \pm 0.47$ | $31.8 \pm 2.38$ |

Table 22: Race prediction on UTKFace with race as protected attribute, across sparsities. **CEAG almost always achieves a** $\max_g \psi_g$ **within the threshold**. It also has the minimum $\max_g \psi_g$ across sparsities.

| Sparsity | Method | Train | | | | Test | | |
|---|---|---|---|---|---|---|---|---|
| | | Accuracy | $\Psi_{\text{PW}}$ | $\max_g \psi_g$ | Tol ($\epsilon$) | Accuracy | $\Psi_{\text{PW}}$ | $\max_g \psi_g$ |
| 85 | NFT | $99.7 \pm 0.02$ | $0.3 \pm 0.16$ | $0.2 \pm 0.13$ | – | $80.6 \pm 0.42$ | $7.6 \pm 1.8$ | $2.7 \pm 0.78$ |
| | NFT + ES | $92.1 \pm 4.2$ | $35.1 \pm 20.55$ | $30.3 \pm 17.82$ | – | $81.1 \pm 0.51$ | $10.7 \pm 2.01$ | $5.2 \pm 2.81$ |
| | EL + RB | $99.7 \pm 0.03$ | $0.4 \pm 0.09$ | $0.2 \pm 0.08$ | – | $80.6 \pm 0.44$ | $9.2 \pm 2.67$ | $2.4 \pm 1.16$ |
| | CEAG | $99.6 \pm 0.05$ | $0.8 \pm 0.14$ | $0.2 \pm 0.05$ | $\leq 0.25\%$ ✓ | $80.3 \pm 0.5$ | $8.4 \pm 2.95$ | $2.0 \pm 0.91$ |
| 90 | NFT | $98.2 \pm 0.08$ | $9.9 \pm 0.82$ | $8.7 \pm 0.82$ | – | $79.5 \pm 0.46$ | $6.1 \pm 1.6$ | $2.2 \pm 0.68$ |
| | NFT + ES | $90.6 \pm 4.72$ | $45.5 \pm 22.51$ | $40.6 \pm 20.13$ | – | $81.0 \pm 0.24$ | $8.0 \pm 4.73$ | $5.3 \pm 4.68$ |
| | EL + RB | $98.3 \pm 0.06$ | $4.4 \pm 0.58$ | $3.6 \pm 0.61$ | – | $79.7 \pm 0.37$ | $8.0 \pm 1.68$ | $1.7 \pm 0.91$ |
| | CEAG | $96.1 \pm 0.11$ | $3.5 \pm 0.51$ | $0.8 \pm 0.26$ | $\leq 1\%$ ✓ | $80.5 \pm 0.25$ | $5.2 \pm 1.86$ | $1.5 \pm 0.2$ |
| 92.5 | NFT | $95.1 \pm 0.36$ | $28.2 \pm 3.49$ | $25.3 \pm 3.35$ | – | $79.4 \pm 0.17$ | $6.1 \pm 3.0$ | $2.5 \pm 1.01$ |
| | NFT + ES | $91.2 \pm 4.29$ | $43.5 \pm 10.64$ | $38.7 \pm 8.87$ | – | $80.2 \pm 0.21$ | $5.7 \pm 2.79$ | $3.5 \pm 1.76$ |
| | EL + RB | $95.4 \pm 0.22$ | $14.6 \pm 1.31$ | $12.5 \pm 1.12$ | – | $78.6 \pm 0.33$ | $9.0 \pm 3.2$ | $2.2 \pm 0.95$ |
| | CEAG | $93.4 \pm 0.31$ | $3.5 \pm 0.86$ | $1.1 \pm 0.36$ | $\leq 1\%$ ✗ | $79.6 \pm 0.4$ | $8.0 \pm 2.59$ | $1.3 \pm 0.43$ |

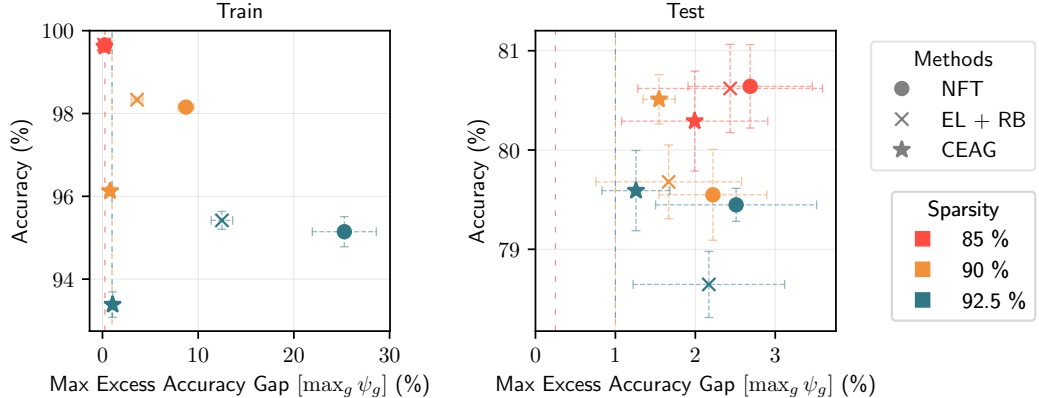

Figure 6: UTKFace race prediction with race as protected attribute.

### F.1.3 INTERSECTIONAL

For the sake of brevity, we use acronyms to refer to the intersectional sub-groups. The acronyms are separated by a dash, the initial part refers to the race and the later part refers to the gender. For instance, **W-M** refers to White and Male. Other races are B-Black, A-Asian, I-Indian, and O-Others.

Table 23: Groupwise train accuracy for race prediction in UTKFace with intersection of race and gender as protected attribute, across sparsities.

| Sparsity | Method | W-M | W-F | B-M | B-F | A-M | A-F | I-M | I-F | O-M | O-F |
|---|---|---|---|---|---|---|---|---|---|---|---|
| 85 | Dense | 99.7 | 99.9 | 99.6 | 99.8 | 99.8 | 99.9 | 99.7 | 99.9 | 99.3 | 99.6 |
| | NFT | $99.6 \pm 0.09$ | $99.7 \pm 0.16$ | $99.6 \pm 0.18$ | $99.7 \pm 0.08$ | $99.6 \pm 0.09$ | $99.9 \pm 0.09$ | $99.7 \pm 0.1$ | $99.8 \pm 0.12$ | $98.6 \pm 0.37$ | $99.2 \pm 0.27$ |
| | EL + RB | $99.5 \pm 0.09$ | $99.5 \pm 0.14$ | $99.6 \pm 0.16$ | $99.7 \pm 0.07$ | $99.7 \pm 0.12$ | $99.9 \pm 0.06$ | $99.7 \pm 0.15$ | $99.9 \pm 0.15$ | $99.6 \pm 0.19$ | $99.7 \pm 0.13$ |
| | CEAG | $99.6 \pm 0.11$ | $99.7 \pm 0.1$ | $99.6 \pm 0.2$ | $99.7 \pm 0.09$ | $99.7 \pm 0.1$ | $100.0 \pm 0.04$ | $99.7 \pm 0.17$ | $99.9 \pm 0.07$ | $99.4 \pm 0.14$ | $99.6 \pm 0.17$ |
| 90 | Dense | 99.7 | 99.9 | 99.6 | 99.8 | 99.8 | 99.9 | 99.7 | 99.9 | 99.3 | 99.6 |
| | NFT | $98.6 \pm 0.2$ | $98.4 \pm 0.21$ | $99.1 \pm 0.27$ | $99.1 \pm 0.23$ | $99.2 \pm 0.24$ | $99.5 \pm 0.15$ | $99.0 \pm 0.33$ | $98.8 \pm 0.21$ | $87.7 \pm 1.07$ | $89.9 \pm 1.88$ |
| | EL + RB | $98.3 \pm 0.31$ | $98.0 \pm 0.29$ | $98.7 \pm 0.44$ | $98.6 \pm 0.28$ | $98.9 \pm 0.27$ | $99.2 \pm 0.15$ | $98.3 \pm 0.6$ | $98.1 \pm 0.45$ | $97.3 \pm 0.64$ | $95.6 \pm 0.59$ |
| | CEAG | $96.4 \pm 0.39$ | $96.0 \pm 0.33$ | $96.1 \pm 0.34$ | $96.2 \pm 0.27$ | $95.3 \pm 0.28$ | $97.0 \pm 0.78$ | $95.7 \pm 0.66$ | $96.2 \pm 0.57$ | $96.0 \pm 0.98$ | $96.6 \pm 1.29$ |
| 92.5 | Dense | 99.7 | 99.9 | 99.6 | 99.8 | 99.8 | 99.9 | 99.7 | 99.9 | 99.3 | 99.6 |
| | NFT | $97.1 \pm 0.61$ | $96.7 \pm 0.42$ | $97.7 \pm 0.46$ | $98.0 \pm 0.27$ | $98.2 \pm 0.33$ | $98.8 \pm 0.34$ | $96.5 \pm 0.54$ | $96.3 \pm 0.71$ | $63.8 \pm 1.49$ | $70.9 \pm 2.09$ |
| | EL + RB | $95.9 \pm 0.48$ | $95.1 \pm 0.43$ | $96.8 \pm 0.32$ | $96.9 \pm 0.23$ | $97.1 \pm 0.5$ | $98.1 \pm 0.58$ | $94.8 \pm 0.83$ | $94.8 \pm 0.51$ | $88.7 \pm 1.91$ | $86.4 \pm 1.64$ |
| | CEAG | $94.2 \pm 0.49$ | $93.4 \pm 0.16$ | $93.6 \pm 0.93$ | $94.2 \pm 0.58$ | $91.9 \pm 1.18$ | $94.2 \pm 0.55$ | $91.6 \pm 0.41$ | $92.8 \pm 1.31$ | $92.0 \pm 0.99$ | $92.7 \pm 1.89$ |

Table 24: Groupwise test accuracy for race prediction in UTKFace with intersection of race and gender as protected attribute, across sparsities.

| Sparsity | Method | W-M | W-F | B-M | B-F | A-M | A-F | I-M | I-F | O-M | O-F |
|---|---|---|---|---|---|---|---|---|---|---|---|
| 85 | Dense | 90.9 | 90.2 | 86.9 | 88.9 | 87.5 | 89.3 | 81.0 | 80.4 | 26.4 | 31.6 |
| | NFT | $87.3 \pm 0.7$ | $86.5 \pm 0.79$ | $82.6 \pm 1.21$ | $86.7 \pm 1.66$ | $84.2 \pm 1.91$ | $87.5 \pm 0.6$ | $74.3 \pm 1.66$ | $78.4 \pm 2.26$ | $29.3 \pm 1.56$ | $32.0 \pm 3.4$ |
| | EL + RB | $87.3 \pm 0.9$ | $86.1 \pm 1.18$ | $82.2 \pm 1.73$ | $85.6 \pm 0.51$ | $84.5 \pm 2.53$ | $86.7 \pm 1.88$ | $74.6 \pm 1.57$ | $78.1 \pm 1.15$ | $31.2 \pm 2.36$ | $32.1 \pm 2.81$ |
| | CEAG | $87.2 \pm 0.65$ | $85.8 \pm 0.76$ | $82.6 \pm 1.27$ | $86.0 \pm 0.96$ | $84.6 \pm 0.52$ | $87.7 \pm 1.76$ | $74.0 \pm 1.18$ | $77.2 \pm 1.7$ | $31.5 \pm 1.83$ | $32.3 \pm 3.18$ |
| 90 | Dense | 90.9 | 90.2 | 86.9 | 88.9 | 87.5 | 89.3 | 81.0 | 80.4 | 26.4 | 31.6 |
| | NFT | $86.8 \pm 0.87$ | $86.4 \pm 0.97$ | $81.9 \pm 1.5$ | $85.2 \pm 1.1$ | $82.2 \pm 2.46$ | $84.4 \pm 1.99$ | $74.2 \pm 0.53$ | $75.4 \pm 1.09$ | $26.8 \pm 3.26$ | $31.6 \pm 1.74$ |
| | EL + RB | $85.9 \pm 1.17$ | $85.6 \pm 1.33$ | $82.5 \pm 0.58$ | $83.8 \pm 0.48$ | $83.4 \pm 1.95$ | $85.0 \pm 2.07$ | $73.8 \pm 1.29$ | $76.1 \pm 1.45$ | $30.3 \pm 1.11$ | $32.3 \pm 2.81$ |
| | CEAG | $86.2 \pm 1.35$ | $87.2 \pm 0.92$ | $83.5 \pm 1.57$ | $85.6 \pm 0.6$ | $82.6 \pm 1.52$ | $86.5 \pm 1.22$ | $76.9 \pm 1.26$ | $76.6 \pm 1.23$ | $25.8 \pm 2.63$ | $29.8 \pm 1.16$ |
| 92.5 | Dense | 90.9 | 90.2 | 86.9 | 88.9 | 87.5 | 89.3 | 81.0 | 80.4 | 26.4 | 31.6 |
| | NFT | $86.5 \pm 1.04$ | $86.6 \pm 0.92$ | $81.9 \pm 1.48$ | $83.0 \pm 1.28$ | $82.3 \pm 1.09$ | $85.5 \pm 1.61$ | $72.8 \pm 1.26$ | $75.6 \pm 2.2$ | $26.8 \pm 3.56$ | $29.6 \pm 1.36$ |
| | EL + RB | $85.0 \pm 0.83$ | $84.9 \pm 0.9$ | $81.8 \pm 0.92$ | $82.8 \pm 1.28$ | $81.7 \pm 0.97$ | $85.6 \pm 1.32$ | $72.8 \pm 0.81$ | $74.0 \pm 2.07$ | $32.8 \pm 6.05$ | $33.9 \pm 2.71$ |
| | CEAG | $86.8 \pm 0.64$ | $85.3 \pm 1.22$ | $82.5 \pm 1.47$ | $84.8 \pm 0.58$ | $81.8 \pm 1.98$ | $86.2 \pm 1.15$ | $73.7 \pm 1.73$ | $76.0 \pm 1.41$ | $29.5 \pm 1.4$ | $30.7 \pm 2.73$ |

Table 25: Race prediction task on the UTKFace dataset with the intersection of race and gender as group attribute, across sparsities. For instance, if a sample has race as *Black* and gender as *Female*, its group label is *Black-Female*. **CEAG consistently achieves a $\max_g \psi_g$ within the threshold, across sparsities.** This table is an extension of Table 2.

| Sparsity | Method | Train | | | | Test | | |
|---|---|---|---|---|---|---|---|---|
| | | Accuracy | $\Psi_{PW}$ | $\max_g \psi_g$ | Tol ($\epsilon$) | Accuracy | $\Psi_{PW}$ | $\max_g \psi_g$ |
| 85 | NFT | $99.6 \pm 0.03$ | $0.8 \pm 0.26$ | $0.5 \pm 0.33$ | – | $80.6 \pm 0.44$ | $10.3 \pm 2.09$ | $3.4 \pm 1.29$ |
| | NFT + ES | $91.8 \pm 4.03$ | $37.0 \pm 21.45$ | $31.8 \pm 18.93$ | – | $81.2 \pm 0.4$ | $13.0 \pm 3.29$ | $5.2 \pm 2.03$ |
| | EL + RB | $99.7 \pm 0.02$ | $0.8 \pm 0.37$ | $0.2 \pm 0.08$ | – | $80.4 \pm 0.29$ | $11.7 \pm 3.4$ | $3.4 \pm 1.53$ |
| | CEAG | $99.7 \pm 0.03$ | $0.5 \pm 0.1$ | $0.2 \pm 0.04$ | $\leq 0.5\%$ ✓ | $80.4 \pm 0.27$ | $12.2 \pm 2.62$ | $3.6 \pm 1.14$ |
| 90 | NFT | $98.1 \pm 0.06$ | $11.5 \pm 0.72$ | $10.0 \pm 0.67$ | – | $79.6 \pm 0.49$ | $8.9 \pm 2.35$ | $3.1 \pm 0.46$ |
| | NFT + ES | $90.5 \pm 4.73$ | $49.8 \pm 23.02$ | $44.8 \pm 20.76$ | – | $81.0 \pm 0.24$ | $12.0 \pm 5.34$ | $6.9 \pm 4.79$ |
| | EL + RB | $98.3 \pm 0.19$ | $3.2 \pm 0.63$ | $2.4 \pm 0.61$ | – | $79.4 \pm 0.5$ | $11.4 \pm 0.91$ | $3.0 \pm 1.06$ |
| | CEAG | $96.2 \pm 0.1$ | $2.4 \pm 0.59$ | $1.0 \pm 0.27$ | $\leq 3\%$ ✓ | $80.2 \pm 0.13$ | $6.0 \pm 2.48$ | $2.3 \pm 1.03$ |
| 92.5 | NFT | $95.1 \pm 0.17$ | $34.2 \pm 1.64$ | $30.7 \pm 1.48$ | – | $79.2 \pm 0.16$ | $8.8 \pm 3.18$ | $3.6 \pm 1.3$ |
| | NFT + ES | $91.2 \pm 2.66$ | $53.3 \pm 9.55$ | $48.0 \pm 8.28$ | – | $80.4 \pm 0.35$ | $7.5 \pm 3.41$ | $5.4 \pm 3.13$ |
| | EL + RB | $95.4 \pm 0.27$ | $11.1 \pm 1.45$ | $8.6 \pm 1.42$ | – | $78.7 \pm 0.27$ | $16.3 \pm 3.92$ | $3.3 \pm 0.62$ |
| | CEAG | $93.4 \pm 0.31$ | $3.8 \pm 0.4$ | $2.3 \pm 0.41$ | $\leq 3\%$ ✓ | $79.5 \pm 0.14$ | $10.8 \pm 2.21$ | $3.3 \pm 1.02$ |

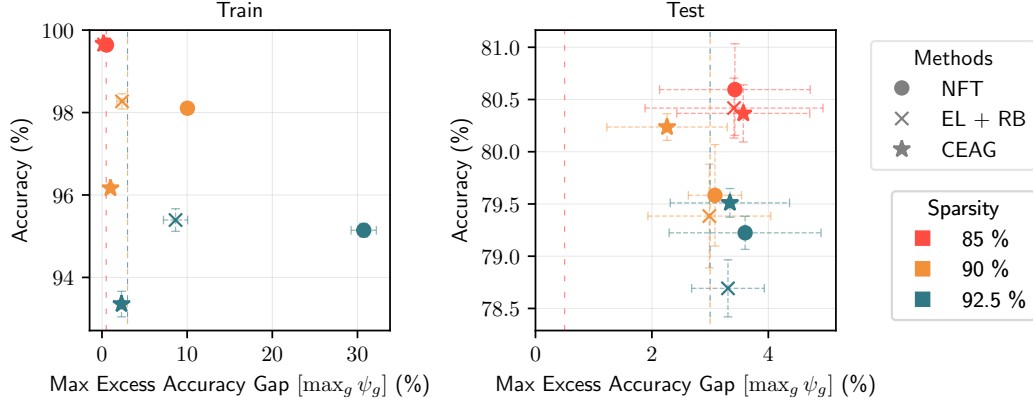

Figure 7: UTKFace race prediction with race and gender (intersectional) as protected attributes.

## F.2   FAIRFACE

We make use of ResNet-34 models (He et al., 2016) on FairFace, similar to Lin et al. (2022).

### F.2.1 GENDER

Table 26: Groupwise train accuracy for gender prediction in FairFace with race as protected attribute.

| Sparsity | Method | East Asian | Indian | Black | White | Middle Eastern | Latino Hispanic | S.E. Asian |
|---|---|---|---|---|---|---|---|---|
| 99 | Dense | 97.2 | 97.1 | 94.9 | 97.4 | 98.0 | 97.2 | 96.8 |
| | NFT | $99.4 \pm 0.21$ | $99.4 \pm 0.23$ | $98.9 \pm 0.12$ | $99.3 \pm 0.14$ | $99.5 \pm 0.15$ | $99.3 \pm 0.2$ | $99.2 \pm 0.31$ |
| | EL + RB | $99.2 \pm 0.21$ | $99.4 \pm 0.11$ | $99.0 \pm 0.24$ | $99.4 \pm 0.03$ | $99.6 \pm 0.06$ | $99.4 \pm 0.09$ | $99.2 \pm 0.13$ |
| | CEAG | $99.3 \pm 0.16$ | $99.2 \pm 0.21$ | $98.9 \pm 0.25$ | $99.3 \pm 0.12$ | $99.5 \pm 0.13$ | $99.3 \pm 0.15$ | $99.1 \pm 0.11$ |

Table 27: Groupwise test accuracy for gender prediction in FairFace with race as protected attribute.

| Sparsity | Method | East Asian | Indian | Black | White | Middle Eastern | Latino Hispanic | S.E. Asian |
|---|---|---|---|---|---|---|---|---|
| 99 | Dense | 95.2 | 95.6 | 90.0 | 94.2 | 96.6 | 95.2 | 94.4 |
| | NFT | $92.1 \pm 0.54$ | $93.4 \pm 0.53$ | $86.5 \pm 0.78$ | $91.6 \pm 0.48$ | $95.1 \pm 0.56$ | $93.4 \pm 0.59$ | $91.4 \pm 0.59$ |
| | EL + RB | $92.4 \pm 0.41$ | $92.9 \pm 1.14$ | $86.8 \pm 0.68$ | $91.8 \pm 0.2$ | $94.9 \pm 0.39$ | $93.6 \pm 0.36$ | $91.4 \pm 0.33$ |
| | CEAG | $92.8 \pm 0.49$ | $92.7 \pm 0.32$ | $86.2 \pm 0.45$ | $91.3 \pm 0.66$ | $94.6 \pm 0.25$ | $93.8 \pm 0.25$ | $91.2 \pm 0.55$ |

Table 28: Gender prediction on FairFace with race as protected attribute. **CEAG achieves a** $\max_g \psi_g$ **within the threshold**.

| Sparsity | Method | Train | | | | Test | | |
|---|---|---|---|---|---|---|---|---|
| | | Accuracy | $\Psi_{\text{PW}}$ | $\max_g \psi_g$ | Tol ($\epsilon$) | Accuracy | $\Psi_{\text{PW}}$ | $\max_g \psi_g$ |
| 99 | NFT | $99.3 \pm 0.17$ | $2.6 \pm 0.23$ | $0.9 \pm 0.13$ | – | $91.8 \pm 0.17$ | $2.4 \pm 0.81$ | $1.1 \pm 0.42$ |
| | NFT + ES | $97.6 \pm 1.29$ | $1.5 \pm 0.67$ | $0.5 \pm 0.19$ | – | $92.2 \pm 0.13$ | $2.8 \pm 0.45$ | $1.5 \pm 0.35$ |
| | EL + RB | $99.3 \pm 0.09$ | $2.7 \pm 0.22$ | $0.8 \pm 0.07$ | – | $91.9 \pm 0.21$ | $2.1 \pm 0.39$ | $1.0 \pm 0.34$ |
| | FairGrape | – | – | – | – | 90.5 | 2.3 | 1.0 |
| | CEAG | $99.2 \pm 0.14$ | $2.7 \pm 0.22$ | $0.9 \pm 0.07$ | $\leq 1\%$ ✓ | $91.7 \pm 0.1$ | $2.4 \pm 0.48$ | $1.2 \pm 0.44$ |

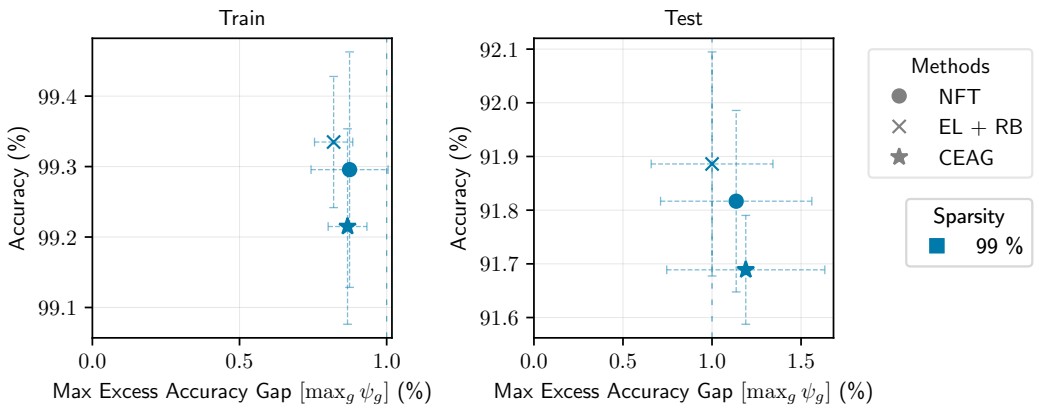

Figure 8: FairFace gender prediction with race as protected attribute.

### F.2.2 RACE

Table 29: Groupwise train accuracy for race prediction in FairFace with race as protected attribute.

| Sparsity | Method | East Asian | Indian | Black | White | Middle Eastern | Latino Hispanic | S.E. Asian |
|---|---|---|---|---|---|---|---|---|
| 99 | Dense | 84.8 | 81.9 | 90.9 | 84.7 | 76.8 | 68.0 | 74.1 |
| | NFT | $81.1 \pm 0.88$ | $78.2 \pm 0.59$ | $87.3 \pm 0.43$ | $80.9 \pm 1.06$ | $70.2 \pm 0.73$ | $63.2 \pm 1.47$ | $68.6 \pm 1.18$ |
| | EL + RB | $78.7 \pm 0.58$ | $77.0 \pm 0.95$ | $84.7 \pm 0.93$ | $78.8 \pm 1.03$ | $71.9 \pm 0.39$ | $69.7 \pm 1.13$ | $70.1 \pm 1.13$ |
| | CEAG | $80.7 \pm 0.95$ | $77.9 \pm 0.61$ | $87.2 \pm 0.61$ | $80.4 \pm 1.14$ | $73.7 \pm 0.79$ | $63.1 \pm 1.46$ | $68.5 \pm 1.22$ |

Table 30: Groupwise test accuracy for race prediction in FairFace with race as metrics attribute.

| Sparsity | Method | East Asian | Indian | Black | White | Middle Eastern | Latino Hispanic | S.E. Asian |
|---|---|---|---|---|---|---|---|---|
| 99 | Dense | 78.3 | 72.2 | 86.2 | 77.2 | 63.9 | 58.1 | 64.3 |
| | NFT | $72.5 \pm 0.98$ | $65.8 \pm 0.97$ | $81.4 \pm 0.41$ | $70.1 \pm 1.34$ | $55.7 \pm 0.94$ | $50.9 \pm 1.19$ | $56.1 \pm 1.06$ |
| | EL + RB | $70.6 \pm 1.28$ | $65.0 \pm 1.24$ | $79.3 \pm 0.57$ | $68.3 \pm 0.85$ | $56.5 \pm 0.54$ | $54.8 \pm 1.07$ | $57.7 \pm 1.86$ |
| | CEAG | $72.5 \pm 1.25$ | $65.8 \pm 1.04$ | $81.5 \pm 0.6$ | $69.5 \pm 1.32$ | $57.5 \pm 0.84$ | $50.3 \pm 1.22$ | $56.4 \pm 0.89$ |

Table 31: Race prediction task on FairFace with race as group attribute. Tol ($\epsilon$) is the tolerance hyper-parameter of CEAG. We do not specify $\epsilon$ for other formulations as they do not admit a tolerance. **CEAG achieves a** $\max_g \psi_g$ **within the threshold**. This table is already presented in the main paper as Table 1, we include this for completeness.

| Sparsity | Method | Train | | | | Test | | |
|---|---|---|---|---|---|---|---|---|
| | | Accuracy | $\Psi_{PW}$ | $\max_g \psi_g$ | Tol ($\epsilon$) | Accuracy | $\Psi_{PW}$ | $\max_g \psi_g$ |
| 99 | NFT | $76.1 \pm 0.19$ | $3.9 \pm 0.91$ | $2.3 \pm 0.26$ | – | $65.2 \pm 0.44$ | $4.2 \pm 0.51$ | $2.1 \pm 0.51$ |
| | NFT + ES | $74.0 \pm 2.55$ | $7.2 \pm 3.35$ | $4.0 \pm 1.38$ | – | $65.4 \pm 0.35$ | $6.3 \pm 2.59$ | $2.9 \pm 1.3$ |
| | EL + RB | $76.1 \pm 0.13$ | $8.8 \pm 1.26$ | $2.6 \pm 0.21$ | – | $65.1 \pm 0.44$ | $6.0 \pm 1.51$ | $2.4 \pm 0.36$ |
| | FairGrape | – | – | – | – | 65.1 | 15.9 | 10.7 |
| | CEAG | $76.2 \pm 0.13$ | $3.5 \pm 0.61$ | $1.8 \pm 0.42$ | $\leq 2\%$ ✓ | $65.2 \pm 0.37$ | $4.3 \pm 0.8$ | $2.0 \pm 0.32$ |

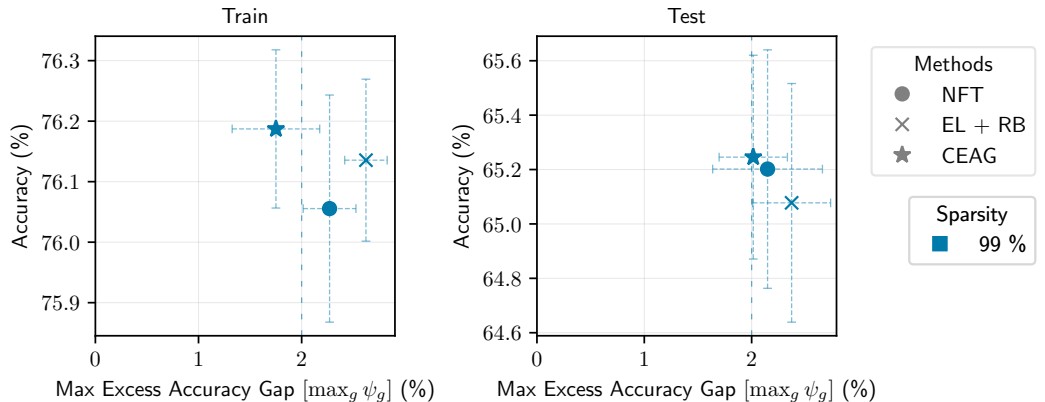

Figure 9: FairFace race prediction with race as protected attribute.

### F.2.3 INTERSECTIONAL

For the sake of brevity, we use acronyms to refer to the intersectional sub-groups. The acronyms are separated by a dash, the initial part refers to the race and the later part refers to the gender. For instance, **EA-M** refers to East Asian and Male. Other races are I-Indian, B-Black, W-White, ME-Middle Eastern, LH-Latino Hispanic, and SA-South East Asian.

Table 32: Groupwise train accuracy for race prediction in FairFace with intersection of race and gender as protected attribute.

| Sparsity | Method | EA-M | EA-F | I-M | I-F | B-M | B-F | W-M | W-F | ME-M | ME-F | LH-M | LH-F | SA-M | SA-F |
|---|---|---|---|---|---|---|---|---|---|---|---|---|---|---|---|
| 99 | Dense | 83.8 | 86.3 | 80.9 | 83.3 | 89.8 | 91.9 | 83.7 | 85.4 | 80.7 | 68.0 | 64.9 | 70.9 | 75.3 | 72.4 |
| | NFT | $78.6 \pm 0.91$ | $83.2 \pm 0.81$ | $76.8 \pm 0.6$ | $78.9 \pm 0.48$ | $86.5 \pm 0.57$ | $87.9 \pm 0.43$ | $79.3 \pm 1.14$ | $82.0 \pm 1.08$ | $74.0 \pm 0.49$ | $60.3 \pm 1.02$ | $59.3 \pm 1.72$ | $66.4 \pm 1.31$ | $69.8 \pm 1.57$ | $67.1 \pm 1.16$ |
| | EL + RB | $77.9 \pm 0.77$ | $82.5 \pm 0.71$ | $76.7 \pm 0.72$ | $78.9 \pm 0.31$ | $86.0 \pm 0.58$ | $87.3 \pm 0.52$ | $78.8 \pm 1.33$ | $81.5 \pm 1.03$ | $74.4 \pm 0.56$ | $62.6 \pm 0.8$ | $61.0 \pm 2.1$ | $67.4 \pm 1.57$ | $70.2 \pm 1.17$ | $67.3 \pm 1.22$ |
| | CEAG | $78.6 \pm 0.72$ | $83.1 \pm 0.85$ | $76.5 \pm 0.77$ | $79.0 \pm 0.35$ | $86.5 \pm 0.58$ | $87.9 \pm 0.57$ | $79.1 \pm 1.15$ | $81.7 \pm 1.13$ | $74.8 \pm 0.77$ | $63.1 \pm 0.75$ | $59.5 \pm 1.59$ | $66.0 \pm 1.31$ | $69.5 \pm 1.48$ | $66.8 \pm 1.16$ |

Table 33: Groupwise test accuracy for race prediction in FairFace with intersection of race and gender as protected attribute.

| Sparsity | Method | EA-M | EA-F | I-M | I-F | B-M | B-F | W-M | W-F | ME-M | ME-F | LH-M | LH-F | SA-M | SA-F |
|---|---|---|---|---|---|---|---|---|---|---|---|---|---|---|---|
| 99 | Dense | 78.8 | 77.9 | 68.5 | 75.9 | 86.1 | 86.4 | 75.6 | 79.1 | 71.7 | 47.7 | 53.5 | 62.5 | 67.3 | 61.0 |
| | NFT | $70.6 \pm 0.98$ | $74.3 \pm 1.63$ | $61.4 \pm 1.42$ | $70.8 \pm 0.9$ | $83.1 \pm 0.23$ | $79.8 \pm 0.53$ | $69.3 \pm 1.33$ | $71.0 \pm 1.78$ | $62.4 \pm 1.23$ | $42.0 \pm 2.72$ | $46.1 \pm 1.42$ | $55.7 \pm 1.28$ | $57.6 \pm 0.83$ | $54.8 \pm 1.0$ |
| | EL + RB | $69.6 \pm 0.95$ | $73.9 \pm 1.73$ | $60.7 \pm 1.62$ | $70.4 \pm 0.64$ | $82.7 \pm 1.02$ | $79.2 \pm 1.19$ | $68.8 \pm 1.3$ | $70.3 \pm 1.2$ | $62.8 \pm 0.85$ | $44.2 \pm 1.93$ | $46.3 \pm 1.69$ | $56.1 \pm 1.42$ | $58.0 \pm 0.72$ | $55.2 \pm 0.84$ |
| | CEAG | $70.7 \pm 1.15$ | $74.3 \pm 1.87$ | $61.2 \pm 1.48$ | $70.9 \pm 1.11$ | $82.5 \pm 0.24$ | $80.1 \pm 0.73$ | $69.4 \pm 1.22$ | $70.5 \pm 1.6$ | $62.6 \pm 1.05$ | $43.3 \pm 1.74$ | $46.1 \pm 1.18$ | $55.1 \pm 2.6$ | $57.6 \pm 0.82$ | $55.1 \pm 1.32$ |

Table 34: Race prediction on FairFace with intersection of gender and race as protected attribute. **CEAG achieves a** $\max_g \psi_g$ **within the threshold**.

| Sparsity | Method | Train | | | | Test | | |
|---|---|---|---|---|---|---|---|---|
| | | Accuracy | $\Psi_{\mathrm{PW}}$ | $\max_g \psi_g$ | Tol ($\epsilon$) | Accuracy | $\Psi_{\mathrm{PW}}$ | $\max_g \psi_g$ |
| 99 | NFT | $75.8 \pm 0.17$ | $5.2 \pm 0.94$ | $2.9 \pm 0.42$ | – | $65.3 \pm 0.45$ | $7.6 \pm 0.66$ | $3.4 \pm 0.73$ |
| | NFT + ES | $73.7 \pm 2.33$ | $8.9 \pm 3.83$ | $4.6 \pm 2.14$ | – | $65.4 \pm 0.37$ | $9.0 \pm 2.45$ | $4.3 \pm 1.56$ |
| | EL + RB | $75.8 \pm 0.16$ | $4.0 \pm 1.72$ | $2.3 \pm 0.15$ | – | $65.1 \pm 0.4$ | $7.6 \pm 0.96$ | $3.1 \pm 0.36$ |
| | CEAG | $75.8 \pm 0.14$ | $4.3 \pm 0.42$ | $2.3 \pm 0.26$ | $\leq 2.5\%$ ✓ | $65.3 \pm 0.51$ | $7.6 \pm 1.23$ | $3.5 \pm 0.97$ |

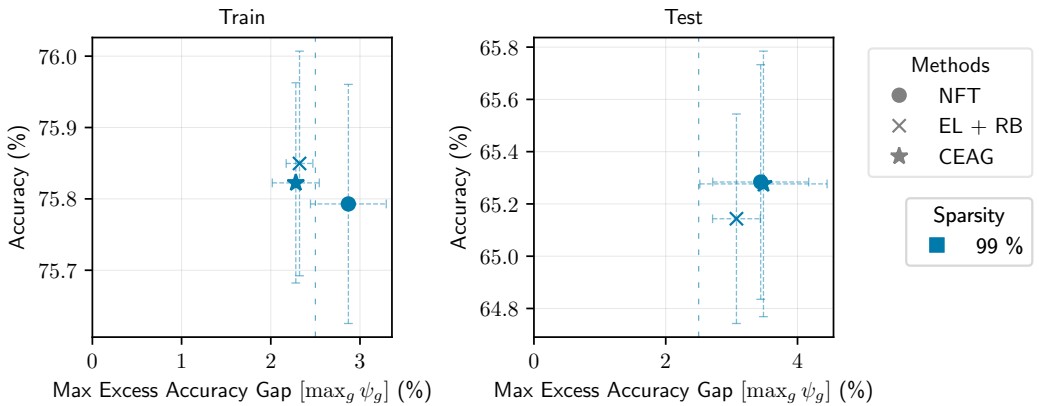

Figure 10: FairFace race prediction with race and gender (intersection) as protected attribute.

## F.3 CIFAR-100

We consider CifarResNet-56 (Chen, 2021) models for this task. We consider a scenario where both the target and group attributes correspond to the class labels. In the context of mitigating the disparate impact of pruning, we want to ensure that none of the classes degrade more than the average degradation with a tolerance of $\epsilon$. The dense model performance is 72.61%.

Table 35: CIFAR-100 classification with the group attributed being the class labels. **CEAG** yields the best model in terms of disparate impact on the training set, and is competitive in terms of $\Psi_{\mathrm{PW}}$ and $\max_g \psi_g$ on the test set.

| Sparsity | Method | Train | | | | Test | | |
|---|---|---|---|---|---|---|---|---|
| | | Accuracy | $\Psi_{\mathrm{PW}}$ | $\max_g \psi_g$ | Tol ($\epsilon$) | Accuracy | $\Psi_{\mathrm{PW}}$ | $\max_g \psi_g$ |
| 90 | NFT | $99.9 \pm 0.0$ | $0.9 \pm 0.18$ | $0.4 \pm 0.18$ | – | $66.7 \pm 0.35$ | $25.0 \pm 3.08$ | $12.9 \pm 2.92$ |
| | NFT + ES | $99.9 \pm 0.01$ | $1.1 \pm 0.3$ | $0.6 \pm 0.3$ | – | $66.9 \pm 0.27$ | $24.6 \pm 1.52$ | $12.7 \pm 1.72$ |
| | EL | $99.8 \pm 0.03$ | $3.1 \pm 0.61$ | $2.4 \pm 0.59$ | – | $67.0 \pm 0.32$ | $24.4 \pm 3.13$ | $12.4 \pm 1.72$ |
| | EL + RB | $99.9 \pm 0.01$ | $1.3 \pm 0.23$ | $0.8 \pm 0.22$ | – | $67.0 \pm 0.38$ | $23.6 \pm 1.52$ | $12.2 \pm 2.06$ |
| | CEAG (no RB) | $100.0 \pm 0.01$ | $1.0 \pm 0.09$ | $0.4 \pm 0.09$ | $\leq 1\%$ ✓ | $66.4 \pm 0.31$ | $26.4 \pm 3.21$ | $13.8 \pm 1.77$ |
| | CEAG | $99.9 \pm 0.01$ | $1.0 \pm 0.09$ | $0.4 \pm 0.08$ | $\leq 1\%$ ✓ | $66.7 \pm 0.34$ | $23.4 \pm 1.52$ | $12.5 \pm 1.4$ |
| 92.5 | NFT | $99.8 \pm 0.02$ | $3.7 \pm 0.86$ | $3.0 \pm 0.87$ | – | $64.9 \pm 0.39$ | $26.2 \pm 5.22$ | $14.3 \pm 3.39$ |
| | NFT + ES | $99.3 \pm 0.2$ | $6.8 \pm 1.9$ | $5.8 \pm 1.8$ | – | $65.2 \pm 0.42$ | $27.4 \pm 2.3$ | $14.6 \pm 1.97$ |
| | EL | $98.5 \pm 0.09$ | $11.3 \pm 0.9$ | $9.8 \pm 0.95$ | – | $65.3 \pm 0.51$ | $25.8 \pm 2.05$ | $14.1 \pm 1.29$ |
| | EL + RB | $99.5 \pm 0.01$ | $6.7 \pm 1.42$ | $5.7 \pm 1.48$ | – | $65.3 \pm 0.41$ | $24.2 \pm 2.86$ | $13.3 \pm 2.35$ |
| | CEAG (no RB) | $99.6 \pm 0.04$ | $2.6 \pm 0.3$ | $1.7 \pm 0.19$ | $\leq 2\%$ ✓ | $65.0 \pm 0.37$ | $27.2 \pm 2.59$ | $14.9 \pm 2.48$ |
| | CEAG | $99.6 \pm 0.04$ | $2.4 \pm 0.17$ | $1.6 \pm 0.15$ | $\leq 2\%$ ✓ | $64.8 \pm 0.3$ | $25.0 \pm 1.87$ | $13.8 \pm 1.16$ |
| 95 | NFT | $96.2 \pm 0.09$ | $14.8 \pm 1.16$ | $11.1 \pm 1.24$ | – | $62.6 \pm 0.29$ | $28.4 \pm 3.21$ | $15.6 \pm 2.57$ |
| | NFT + ES | $93.9 \pm 0.83$ | $20.0 \pm 1.52$ | $14.6 \pm 1.43$ | – | $63.3 \pm 0.21$ | $30.6 \pm 5.55$ | $16.1 \pm 3.22$ |
| | EL | $87.9 \pm 0.45$ | $21.4 \pm 1.36$ | $14.5 \pm 1.31$ | – | $60.7 \pm 0.48$ | $32.8 \pm 3.7$ | $16.1 \pm 4.57$ |
| | EL + RB | $94.6 \pm 0.12$ | $18.4 \pm 1.05$ | $13.6 \pm 1.07$ | – | $62.8 \pm 0.16$ | $27.4 \pm 1.34$ | $14.8 \pm 0.85$ |
| | CEAG (no RB) | $95.8 \pm 0.15$ | $9.2 \pm 0.52$ | $5.8 \pm 0.53$ | $\leq 5\%$ ✗ | $62.5 \pm 0.41$ | $29.6 \pm 4.34$ | $17.1 \pm 3.59$ |
| | CEAG | $95.6 \pm 0.12$ | $9.1 \pm 0.64$ | $5.7 \pm 0.49$ | $\leq 5\%$ ✗ | $62.7 \pm 0.28$ | $27.6 \pm 1.14$ | $14.8 \pm 1.52$ |

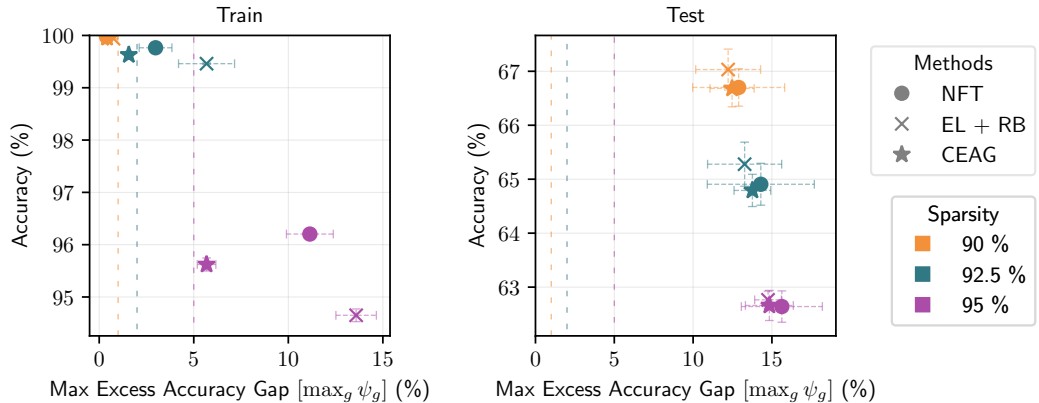

Figure 11: CIFAR-100 classification with both target and protected attribute being the class labels.

