# OpenReview forum: "Balancing Act: Constraining Disparate Impact in Sparse Models"
_ICLR.cc/2024/Conference — ICLR 2024 poster_

### Official Review · Reviewer_qBid · 2023-10-26

**Soundness:** 3 good
**Presentation:** 3 good
**Contribution:** 3 good
**Rating:** 6
**Confidence:** 3

**Summary:**

This paper proposes a method for mitigating disparate effect on model accuracy by formulating the task as a constrained optimization problem and solving it via alternative gradient descent. The introduced approach manages to decrease the gap between average model performance and worst performance on a subgroup of data while preserving the mean accuracy on the target task.

**Strengths:**

The problem being addressed is quite novel in sparsification community and is of significant interest to practitioners, especially in safety-critical applications. The approach looks very reasonable and directly optimizes the imposed constraints via solving a min-max problem. The method is quite non-trivial and comprises several interesting ideas - alternating between non-differentiable accuracy constrain term and excess loss term, as well as use of replay buffers for stabilization of optimization.

CEAG outperforms existing alternatives on a couple of benchmarks - UTKFace and CIFAR100. The method doesn’t add significant computation overhead compared to the standard training procedure.

**Weaknesses:**

While the proposed method manages to keep train accuracy on subgroups within desired tolerance bounds, seems it has hard to achieve on the test data, especially in the setting with many classes and subgroups (for example in the provided CIFAR-100 experiment).

The difference between $\mathrm{max}_g \psi_g $ on the hold-out-data between NFT and CEAG doesn’t seem to be very significant in many cases - 2.0 vs 2.1 on 99% sparsity UTKFace *(Table1)*, 3.3 vs 3.6 for 92.5% *(Table2)*, 13.8 vs 14.3 on CIFAR-100 *(Table3)*. Given the standard deviation of the runs, improvement of CEAG appears to be statistically insignificant.

Experimental validation is not exhaustive enough. To demonstrate the efficiency in a more large scale and practically relevant scenario one could consider more diverse and large-scale dataset, such as ImageNet (or ImageNet-LT version), or one of the iNaturalist versions, considering only large hierarchy groups to make the task computationally tangible.

*Minor*. The pruning strategy called in the paper is customary named **Gradual Magnitude Pruning** (GMP) [2], where after each pruning step one continues training the model from the current state. **Iterative Magnitude Pruning** (IMP) [3] adopted in discovery of Lottery Tickets rewinds the weights to initialization after pruning step. I would suggest calling the method GMP to avoid confusion.

---
[1] Liu, Ziwei, et al. "Large-scale long-tailed recognition in an open world." Proceedings of the IEEE/CVF conference on computer vision and pattern recognition. 2019.

[2] Zhu, Michael, and Suyog Gupta. "To prune, or not to prune: exploring the efficacy of pruning for model compression." arXiv preprint arXiv:1710.01878 (2017).

[3] Frankle, Jonathan, and Michael Carbin. "The lottery ticket hypothesis: Finding sparse, trainable neural networks." arXiv preprint arXiv:1803.03635 (2018).

**Questions:**

How sensitive is the algorithm to the initialization of dual parameters $\lambda_g$ and the corresponding update rule?

---

> ### Author Response · Authors · 2023-11-16
>
> We would like to thank the reviewer for acknowledging the novelty of our method and the importance of the research to the community.
>
> ---
>
> > While the proposed method manages to keep train accuracy on subgroups within desired tolerance bounds, seems it has hard to achieve on the test data, especially in the setting with many classes and subgroups (for example in the provided CIFAR-100 experiment).
>
>
> We agree with the reviewer's observation about the issue of generalization to the test data. In fact, we had highlighted this issue explicitly in the final paragraph of the introduction. We have rephrased our statement to make it clearer:
>
> > Our experimental results indicate that _all methods considered in this paper (including ours) fail to mitigate pruning-induced disparities on unseen data_. To the best of our knowledge, we are the first to document this generalization challenge. Despite this, our proposed method constitutes a step in the right direction since our approach is _the only one_ that reliably mitigates the disparate impact of pruning on the training set. We hope our empirical observations will motivate further research on improving the generalization properties of methods for mitigating the disparate impact of pruning.
>
> We hypothesize this behavior was not observed before due to:
> * the previous papers did not report the performance of the model in training, and
> * they were relying on heuristic metrics for quantifying disparate impact (for example, measuring the standard deviation of the group accuracy gaps $  std_{g \in \mathcal{G}} (\Delta_g  )$  ), rather than measuring  $\Psi_{\text{PW}}$ directly.
>
> We completely agree with the reviewer on the importance of tackling the issue of generalization for disparity-mitigation techniques. We would like to highlight the role our paper has in documenting the generalization issue as a main challenge for the field.
>
> ---
>
> >The difference between $\max_g \Psi_g$ on the hold-out-data between NFT and CEAG doesn’t seem to be very significant in many cases - 2.0 vs 2.1 on 99% sparsity UTKFace (Table1), 3.3 vs 3.6 for 92.5% (Table2), 13.8 vs 14.3 on CIFAR-100 (Table3). Given the standard deviation of the runs, improvement of CEAG appears to be statistically insignificant.
>
> As mentioned before, all considered methods, including ours, struggle to mitigate disparity on the test data. In that sense, we agree with the reviewer's claim about the lack of statistical significance of our improvements **in the test set**.
>
> However, despite the challenges on the test set, we would like to emphasize that, all else being equal, our method should be preferred to existing alternatives since it is **the only one to reliably mitigate the disparate impact on the training set**.

---

> ### Author Response · Authors · 2023-11-16
>
> > Experimental validation is not exhaustive enough. To demonstrate the efficiency in a more large-scale and practically relevant scenario one could consider more diverse and large-scale dataset, such as ImageNet (or ImageNet-LT version), or one of the iNaturalist versions, considering only large hierarchy groups to make the task computationally tangible.
>
> We respectfully disagree with the reviewer's assessment of the scale and thoroughness of our experimental results.
>
> To the best of our knowledge, existing papers in the literature about *mitigating* the disparate impact of pruning, focus on the UTKFace, FairFace, and CelebA datasets. Besides UTKFace, Tran et al. [A] carry out additional experiments on small-scale datasets such as MNIST or SVHN. Hooker et al. [B] consider the ImageNet dataset, but *limit themselves to documenting the existence of pruning-induced disparate impact*, and do not propose a mitigation scheme.
>
> Our paper contains experiments on UTKFace and FairFace, as in existing works. Additionally, we executed experiments for these datasets on the "intersectional setting", leading to a larger number (2x) of protected groups than considered in previous papers. **The intersectional setting was not previously considered in this literature**.
>
> We decided against the CelebA task in Lin et al. [C] since it is a multi-attribute prediction task, which could make the assessment of disparate impact challenging (since overall accuracy is aggregated across prediction attributes).
>
> Importantly, we introduced the CIFAR100 task to demonstrate the *scalability of our method to hundreds of constraints*. This is an important contribution taking into account that prior work explored tasks with only up to 10 protected groups. Moreover, existing methods such as FairGRAPE [C] require prohibitive costs in terms of computation and storage, increasing with the number of constraints. In Appendix D2 "Additional Experiments > Computational Overhead" (in revised version), we provide time measurements which demonstrate that our method seamlessly scales to hundreds of constraints.
>
> We understand the reviewer's interest in testing our method on ever-larger tasks such as ImageNet or iNaturalist. We would like to highlight that our method can be directly applied to these larger tasks without significant overhead. However, we believe that our experimental validation is more than sufficient *in the context of the existing literature*.
>
> Finally, **Reviewer KSs2** requested experiments on structured sparsity. This setting is practically relevant (as **Reviewer qBid** requested) since it yields sparsity patterns that can be leveraged for fast computation in existing hardware (unlike unstructured sparsity). Arguably, good performance of a mitigation method on both the structured and unstructured settings is more indicative of the method's benefits than testing it on a new dataset.
> **Please see additional experiments** executed during the discussion phase using structured sparsity in our response to **Reviewer KSs2**.
>
>
> [A] Cuong Tran, Ferdinando Fioretto, Jung-Eun Kim, and Rakshit Naidu. Pruning has a disparate impact on model accuracy. In NeurIPS, 2022.
>
> [B] Sara Hooker, Nyalleng Moorosi, Gregory Clark, Samy Bengio, and Emily Denton. Characterising Bias in Compressed Models. arXiv:2010.03058, 2020.
>
> [C] Xiaofeng Lin, Seungbae Kim, and Jungseock Joo. FairGRAPE: Fairness-Aware GRAdient Pruning mEthod for Face Attribute Classification. In ECCV, 2022.
>
> ---
>
> > Minor. The pruning strategy called in the paper is customary named Gradual Magnitude Pruning (GMP) [2], where after each pruning step one continues training the model from the current state. Iterative Magnitude Pruning (IMP) [3] adopted in discovery of Lottery Tickets rewinds the weights to initialization after pruning step. I would suggest calling the method GMP to avoid confusion.
>
> We thank the reviewer for the suggestion. We will adjust the paper and refer to the method proposed by Zhu and Gupta as Gradual Magnitude Pruning (GMP) to avoid confusion with Iterative Magnitude Pruning (IMP) popularized by Frankle and Carbin.

---

> > ### Comment · Reviewer_qBid · 2023-11-17
> >
> > Thanks for your answer and clarifications.
> > Now I understand that the work has significant phenomenological contribution in addition to the introduction of a new method. Structured sparsity experiments provide an additional validation of the CEAG approach and contain several interesting findings.
> > While from the practical point of view one is interested in preserving the accuracy uniformly across the **unseen**, **hold-out** data rather than the train set, I think that the obtained results are valuable for the research community.
> >
> > I agree, that the evaluation presented in the work follows the previous work and introduces some new results. However, since the problem considered is quite generic, potentially arising in many real-world scenarios, I think an experiment on a large-scale dataset would significantly strengthen the experimental section of the paper.
> >
> > Nevertheless, I decide to raise my score after reading responses addressed to me and other reviewers.

---

### Official Review · Reviewer_KSs2 · 2023-11-01

**Soundness:** 3 good
**Presentation:** 3 good
**Contribution:** 3 good
**Rating:** 8
**Confidence:** 3

**Summary:**

This work proposes a new approach for fine-tuning sparse models after pruning. The proposed approach involves formulating a constrained problem, where loss function is optimized subject to a group-wise accuracy constraint (that is, the accuracy drop for each group should be bounded by a tolerance $\varepsilon$). This Lagrangian of this constrained problem is then optimized with standard gradient-based methods. This approach is empirically shown to reliably generate models where the differences in the group-wise impact of sparsity on accuracy is minimized.

**Strengths:**

The paper has several strengths:

* The proposed approach is interesting, novel, and flexible, and reliably reduces group wise disparities in accuracy after pruning.
* The empirical evaluation is very thorough.
* Limitations of the proposed methods and ethical considerations are discussed thoroughly as well.
* In the appendix, variations of the main results are discussed as well.;/
* The writing and presentation is clear.

**Weaknesses:**

The paper has a few weaknesses.

* This method is independent of the choice of pruning strategy. However, it would have been nice to see the experiments replicated for other pruning strategies, including structured pruning methods.
* A more detailed discussion on the feasibility of the constrained problem given in equation (4) would have been useful for readers.

**Questions:**

* Have the authors tried applying this technique to other pruning methods? For instance, how well would this fine-tuning method work if structured pruning was used instead of unstructured pruning? How well would the method work if other unstructured pruning methods were used instead of IMP (i.e. SynFlow [1] or SNIP [2])?
* Is there a sparsity level at which the method fails to achieve models with the desired worst-case groupwise accuracy loss? Put another way, is there a sparsity level at which the constrained optimization problem described in eq. (5) become infeasible? Can the authors comment on how this might play out in the case of structured pruning?
* Are there any formal results that can be provided for solving the CEAG (eq (5))? For instance, is Algorithm 1 guaranteed to find a solution provided the feasible set is nonempty?
* Have the authors considered ways by which the test-case performance can be improved, say by dataset splits?
* Is the CEAG method affected by dataset imbalance? Suppose certain groups have comparatively fewer samples in the dataset. How, if at all, would this affect the efficacy of the method?

[1] "Pruning neural networks without any data by iteratively conserving synaptic flow", Tanaka et al, 2020.

[2] "SNIP: Single-shot Network Pruning based on Connection Sensitivity" Lee et al, 2019.

---

> ### Author Response · Authors · 2023-11-16
>
> We would like to thank the reviewer for highlighting the importance of the research and the thoroughness and novelty of our paper. We would also like to thank the reviewer for their thorough review and insightful questions about our work.
>
> ---
> > Is there a sparsity level at which the method fails to achieve models with the desired worst-case groupwise accuracy loss? Put another way, is there a sparsity level at which the constrained optimization problem described in eq. (5) become infeasible? Can the authors comment on how this might play out in the case of structured pruning?
>
> It has been previously documented [A, B, C] that deep neural networks are robust to high levels of sparsity, but their training dynamics become challenging. Our constrained formulation suffers from similar challenges, and we would expect that these difficulties become more pronounced in the constrained setting—as per the reviewer's intuition.
>
> For our experiments, we chose commonly used sparsity levels for the different architectures and we were able to find feasible solutions that achieved disparity below that of the (unconstrained) NFT baseline. Empirically, the "breaking point" for the tolerance level did not increase as a result of adding the constraints.
>
> Note that when increasing the sparsity level, we sometimes had to increase the constraint level (e.g. Table 25 in revision [UTKFace > Intersectional]). However, this is mostly due to the loss of model capacity, as is observed in the increase in $\max_g \psi_g$ experienced by the *unconstrained* NFT baseline.
>
>
> [A] Song Han, Jeff Pool, John Tran, William J. Dally. Learning Both Weights and Connections for Efficient Neural Network. In NeurIPS, 2015.
>
> [B] Trevor Gale, Erich Elsen, and Sara Hooker. The State of Sparsity in Deep Neural Networks. arXiv:1902.09574, 2019.
>
> [C] Utku Evci, Fabian Pedregosa, Aidan Gomez, and Erich Elsen. The Difficulty of Training Sparse Neural Networks." arXiv:1906.10732, 2019.
>
> ---
> >Are there any formal results that can be provided for solving the CEAG (eq (5))? For instance, is Algorithm 1 guaranteed to find a solution provided the feasible set is nonempty?
>
> No, even if the feasible set is non-empty, there is no guarantee that Algorithm 1 will find a solution. The possibility for analysis is limited as we consider a non-convex, non-differentiable constrained optimization problem. Viewed from the min-max angle, the use of proxy-constraints results in a non-zero-sum game. Providing convergence guarantees for these optimization problems is a challenging, open area of research, beyond the scope of our work.
>
> We invite the reviewer to read our response to **Reviewer Cvd6**, including some pointers to local convergence results under idealized problem conditions.
>
> ---
>
> > Have the authors considered ways by which the test-case performance can be improved, say by dataset splits?
>
> This is an interesting idea. In the development of the paper, we considered an early-stopping approach based on constraint and objective estimates on a validation set. Due to the reasons detailed below, we decided not to proceed with this approach in our main experiments. However, exploring techniques to improve test-case performance for disparity-mitigation methods is an important research direction.
>
> The early-stopping approach mentioned above required creating validation splits, which were unfortunately not available for the datasets we considered. In order to make fair comparisons with existing methods, we decided to retain the same choice of model architecture and train/test split. However, in some tasks, we make use of models that have been pre-trained on the entire training set (containing our self-made validation datapoints). Thus the constraint and objective estimates on the validation set would be unreliable due to data leakage. Since these challenges would have prevented us from executing a proper ablation study on the benefits of this technique, we decided not to pursue this.

---

> ### Author Response · Authors · 2023-11-16
>
> > Is the CEAG method affected by dataset imbalance? Suppose certain groups have comparatively fewer samples in the dataset. How, if at all, would this affect the efficacy of the method?
>
> This is an important question and of great importance in fairness. Our proposed CEAG method is designed specifically with this issue in mind.
>
> Data imbalance affects training of fairness-aware methods in two main ways:
> * Under-represented groups might not have enough samples to sway the model's updates toward satisfying their constraints.
> * Estimating the constraints associated with under-represented groups may be challenging due to the low number of samples in the stochastic estimates.
>
> CEAG directly addressed both of these challenges by (1) introducing explicit per-group constraints and (2) using Replay Buffers (RB, Section 4.2) to reduce estimation noise.
>
> * If a constraint for any group (under-represented or not) is violated persistently, its associated Lagrange multiplier will increase, biasing the model gradient towards achieving the satisfaction of the constraint.
> * On the other hand, RBs allow our method to scale to a large number of constraints (even in the data imbalance setting) by constructing less noisy estimates of the constraint violations. Moreover, RBs also improve the stability of the training dynamics under an imbalanced dataset with few protected groups.
>
> We would like to highlight that the use of RBs was one of the key components for improving the scalability and stability of Equalized Loss, originally proposed by Tran et al. [A].
>
> Experimentally, our method excels in tasks with high data imbalance (such as intersectional fairness) compared to all other techniques—see Table 2 at 92.5% sparsity.
>
> [A] Cuong Tran, Ferdinando Fioretto, Jung-Eun Kim, and Rakshit Naidu. Pruning has a disparate impact on model accuracy. In NeurIPS, 2022.

---

> ### Author Response · Authors · 2023-11-16
> **Structured Sparsity Experiments**
>
> > Have the authors tried applying this technique to other pruning methods? For instance, how well would this fine-tuning method work if structured pruning was used instead of unstructured pruning? How well would the method work if other unstructured pruning methods were used instead of IMP (i.e. SynFlow [1] or SNIP [2])?
>
> As the reviewer acknowledged, our proposed method is agnostic to the choice of pruning method. We expect it to behave well under popular pruning techniques like those mentioned by the reviewer.
>
> In particular, structured sparsity is practically relevant since it enables fast inference on existing hardware. Following the reviewer's suggestion, **we conducted additional experiments** demonstrating the behavior of our proposed technique (and all other baselines) in this setting.
>
> **Experiment set-up**
> - UTKFace dataset with target=race and protected attribute=race.
> - Same experimental setting as other UTKFace experiments in the main paper.
> - Metrics averaged over 5 seeds.
> - Layerwise structured output-channel sparsity at 85%.
>
> **Results**
> *Please read the clarifications below the table to adequately interpret the behavior of $\Psi_{\text{PW}}$ in test for EL+RB and CEAG!*
>
> | Sparsity |  Method  |            |        Train         |                 |                  |           |        Test        |                 |
> |----------|----------|------------|----------------------|-----------------|------------------|-----------|--------------------|-----------------|
> |          |          |  Accuracy  | $\Psi_{\text{PW}}$   | $\max_g \psi_g$ | Tol ($\epsilon$) | Accuracy  | $\Psi_{\text{PW}}$ | $\max_g \psi_g$ |
> |85        | NFT      | 89.1±0.39  |      50.6±1.08       |    45.8±0.82    |        –         | 80.7±0.27 |       4.1±1.7      |    2.0±0.61     |
> |85        | NFT + ES | 87.7±0.71  |      58.3±3.08       |    52.7±2.91    |        –         | 81.5±0.11 |       4.7±2.06     |    3.2±1.52     |
> |85        | EL + RB  | 86.9±0.47  |      3.7±1.38        |    1.2±0.53     |        –         | 77.9±0.44 |       27.3±2.58    |    3.2±0.65     |
> |85        | CEAG     | 85.6±0.47  |      2.7±0.71        |    1.0±0.36     |        ≤ 1%      | 78.2±0.39 |       30.1±2.19    |    3.7±0.28     |
>
>
>
>
> ***Key-points:***
> * CEAG and EL+RB successfully mitigate pruning-induced disparate impact in the structured sparsity setting.
> * The results show that our method CEAG achieves the lowest disparate impact in train.
> * Our method has a high $\Psi_{\text{PW}}$ in test because the under-represented group *improved* its test accuracy.
>
> ***Clarifications:***
> We note an unusual pattern in the train-vs-test performance between mitigation methods like EL+RB and CEAG, and the NFT baselines: mitigation methods have very low disparity ($\Psi_{\text{PW}}$) in training but very high disparity in test (and vice versa for NFT).
>
> The reason behind this behavior is the performance of the different models on the under-represented group *"Others"*, corresponding to ~3% of the dataset. The relevant metrics (averaged over 5 seeds) are summarized in the table below.
>
> * The pre-trained model had overfitted to the training samples of group *"Others"*.
> * In the case of NFT, the training objective allows the models to sacrifice performance on under-represented classes in order to favor more numerous ones. In particular, we observed that NFT caused a hyper-degradation (EAG) in group *"Others"* of 45.77%.
> * On the other hand, mitigation techniques like EL+RB and CEAG do **not** yield models that obliviously sacrifice the performance on group *"Others"*. For example, for CEAG the loss in performance in group *"Others"* (13.4%) in training is similar to the overall training accuracy reduction (14.14%).
>
>
> In other words, **the mitigation methods successfully prevent the appearance of excessive disparity due to pruning** by making sure no class (under-represented or not) is degraded more than $\epsilon$ beyond the average model degradation.
>
>
> | Metric (avg. over 5 seeds) | Dense | NFT | EL+RB | CEAG |
> | -------- | -------- | -------- | -------- | -------- |
> | Overall Acc. Train | 99.74 | 89.1  |86.9  | 85.6 |
> | Overall Acc. Test  | 83.82 | 80.7 | 77.9 | 78.2 |
> | Acc. Others Train | 99.37 |42.96  | 88.37 | 85.97 |
> | Acc. Others Test | 29.23 | 26.31 | 47.38 | 49.97 |
> | Overall Acc. Gap Train ($\Delta$) | 0 | 10.64 | 12.84 | 14.14 |
> | Overall Acc. Gap Test ($\Delta$) |  0   | 3.12 | 5.92 | 5.62 |
> | Acc. Gap Others Train ($\Delta_{\text{Others}}$) | 0 | 56.41 | 11.0 | 13.4 |
> | Acc. Gap Others Test ($\Delta_{\text{Others}}$) | 0 | 2.92 | -18.15 | -20.74 |
> | EAG Others Train ($\psi_{\text{Others}}$) | 0 | 45.77 ⚠️ | -1.84 | -0.74 |
> | EAG Others Test ($\psi_{\text{Others}}$) | 0 | -0.2 | -24.07 | -26.36 |

---

> > ### Comment · Reviewer_KSs2 · 2023-11-18
> > **Response to Authors**
> >
> > I would like to thank the authors for their very detailed and clear response - you have answered all my questions. However, at this time, I would like to keep my score at 8.

---

### Official Review · Reviewer_Haeo · 2023-11-03

**Soundness:** 3 good
**Presentation:** 3 good
**Contribution:** 3 good
**Rating:** 6
**Confidence:** 4

**Summary:**

This paper provides a new training protocol to manage pruning induced bias. The main aim is to create algorithms in which the difference in accuracy between any two groups is minimised. The authors highlight that their main focus is lessen the effects of compression and thus they treat dense models as baseline.

**Strengths:**

This is a useful research area. With concerns of AI inclusion and climate change, sparse models are very appealing. The authors provide a method to ameliorate some of the effects of compression by limiting the disparity between group performance.
This is a well written paper and the methodology is clear. Implementation and results are well described and the examples in the appendix provide further grounding on their work.

**Weaknesses:**

It is not clear from the analysis provided that solutions always exist given the constraints. Perhaps the authors could more light on this. Is there a relationship between the starting point and how tight the constraints can be?

**Questions:**

Please see weaknesses.

---

> ### Author Response · Authors · 2023-11-16
>
> We thank the reviewer for highlighting the effectiveness of our method and the usefulness of this research to the community.
>
> > It is not clear from the analysis provided that solutions always exist given the constraints. Perhaps the authors could more light on this.
>
> The reviewer is correct. The constrained optimization problem in Eq(5) might be infeasible for certain values of the constraint level $\epsilon$.
> Note that this potential infeasibility depends not only on $\epsilon$ but also on the data distribution, the model capacity and the sparsity level.
>
> Setting $\epsilon$ large enough would yield a problem with a non-empty feasible region. However, this may come at the cost of not enforcing the disparity constraints. (For example, asking for the excess accuracy gaps to be less than 2.)
>
> In the non-convex constrained optimization regime, even if the feasible set is non-empty (and thus a locally optimal solution exists), there is no guarantee that gradient-based methods can converge to it starting from arbitrary initializations.
>
> We would like to highlight that in practice, we have observed that our method reliably achieves feasible solutions across several datasets, model architectures, and sparsity levels. (The constraint levels were set by asking for lower disparity than observed in the baseline NFT method).  Of course, this practical behavior is not a guarantee on the feasibility of the problem, but it indicates there might be a special structure in the feasible set that makes it amenable to optimization with stochastic gradient-based schemes.
>
>
> ---
>
> > Is there a relationship between the starting point and how tight the constraints can be?
>
> We would like to ask the reviewer to rephrase their question since we are not sure we are interpreting it correctly. Our two current interpretations are:
>
>
> [Interp. 1] Is there relationship between the starting point ($\theta_0$, $\lambda_0$) and the lowest constraint level $\epsilon$ that is achievable?
>
> [Interp. 2] Suppose we are given a fixed constraint level $\epsilon$, and suppose one were to initialize the algorithm ($\theta_0$, $\lambda_0$) close enough to being feasible for the constraint level $\epsilon$. Is there a relationship between initial closeness to being feasible vs whether the algorithm converges to a feasible point?

---

> > ### Comment · Reviewer_Haeo · 2023-11-21
> > **Is there a relationship between the starting point and how tight the constraints can be?**
> >
> > Interpretation 1

---

> > > ### Author Response · Authors · 2023-11-22
> > >
> > > No. The set of feasible solutions to the constrained optimization problem that we formulate is independent of the algorithm used to solve said optimization problem (and in particular, of its initialization $\theta_0, \lambda_0$).
> > >
> > > Nonetheless, one might still consider the relationship between the initialization point and how tight the recovered constraints may be in practice. However, It is not really possible to say anything about this connection given the complexity of the feasible set (since the problem we formulate has non-convex and stochastic constraints).
> > >
> > > Moreover, note that the initialization point of the model $\theta_0$ is a result of the pruning process applied to the pre-trained dense model, and is thus independent of our optimization technique. Said pruning process could yield models far or close to being feasible.  The multipliers ($\lambda_0$) are conventionally set to zero. This choice is motivated by the fact that, when a constraint is satisfied, the optimal value for its corresponding Lagrange multiplier is zero.
> > >
> > > Additionally, we would like to highlight that all our experiments are conducted using multiple random seeds, resulting in different model initializations. This deliberate choice allows us to assess the robustness of our proposed method across various starting points. Our results consistently demonstrate the ability of our approach to converge to feasible solutions across diverse initializations, further emphasizing the efficacy and reliability of our method.

---

> ### Author Response · Authors · 2023-11-20
>
> As the end of the discussion period is coming, we wanted to follow up on our clarification request regarding question:
> > Is there a relationship between the starting point and how tight the constraints can be?
>
> We would also like to ask the reviewer whether there are additional concerns they would like to bring to our attention?

---

### Official Review · Reviewer_cVd6 · 2023-11-11

**Soundness:** 3 good
**Presentation:** 3 good
**Contribution:** 3 good
**Rating:** 5
**Confidence:** 2

**Summary:**

This paper deals with the problem of pruning models with desire of not decreasing performance on subsets via constraints.

**Strengths:**

The technique is solid and the experiments are sound.

**Weaknesses:**

Can the paper deals with other pruning techniques to demonstrate the effectiveness of constraint to subsets?

**Questions:**

Can we give some formal results with Equation(6) like the analytical solution or convergenece guarantee?

---

> ### Author Response · Authors · 2023-11-16
>
> We would like to thank the reviewer for highlighting the effectiveness of our techniques and the soundness of our experiments.
>
> > Can the paper deals with other pruning techniques to demonstrate the effectiveness of constraint to subsets?
>
> Yes, during the rebuttal phase, we have carried out additional experiments under the _structured_ sparsity setting. At a high level, we observed similar trends as in the unstructured pruning setting: our method reliably mitigates the disparate impact of pruning in training, while preserving the overall performance of the model. We invite the reviewer to see our response to **Reviewer KSs2** containing the details and results for these experiments.
>
> The additional results on structured sparsity demonstrate that our technique is agnostic and robust to the choice of pruning method. Therefore, we expect it to perform well under other commonly used sparsity techniques.
>
> ----
>
> > Can we give some formal results with Equation(6) like the analytical solution or convergence guarantee?
>
> In general, the min-max problem in Eq(6) does not have an analytical solution, as is the case for many optimization problems considered in the deep learning field.
>
> Under idealized circumstances, it is possible to develop convergence guarantees of gradient-based optimization methods for min-max problems. We have already provided pointers to relevant literature on the related work section and Appendix A.
>
> Unfortunately, the existing literature focuses on proving local convergence of gradient-based methods. This analysis relies on the assumption of an initialization close enough to a solution to the min-max problem, which is unrealistic in our specific problem.
>
> Despite the lack of theoretical guarantees, our experimental results clearly demonstrate that we are able to reliably find feasible solutions (i.e. low disparate impact) that are approximately optimal (i.e. model performance close to the dense baseline).
>
> For the reviewer's convenience, we list the mentioned references here:
>
> [A] Tianyi Lin, Chi Jin, and Michael Jordan. On Gradient Descent Ascent for Nonconvex-Concave Minimax Problems. In ICML, 2020.
>
> [B] Guodong Zhang, Yuanhao Wang, Laurent Lessard, and Roger B Grosse. Near-optimal Local Convergence of Alternating Gradient Descent-Ascent for Minimax Optimization. In AISTATS, 2022.

---

> ### Author Response · Authors · 2023-11-20
>
> As the end of the discussion period is coming, we wanted to know whether we have addressed all of the reviewer’s concerns and whether there are additional changes that the reviewer would like us to bring to the paper?

---

### Author Response · Authors · 2023-11-16
**Official Response to All Reviews**

We thank the reviewers for their time in preparing their reviews. We have attempted to address all your concerns, and highlight the main contents of our response below:

- Additional experiments on structured pruning [**Reviewers KSs2, qBid, cVd6**]
- Clearer statement on generalization issues [**Reviewer qBid**]
- Discussion on the sufficiency of our experiments in the context of existing literature [**Reviewer qBid**]
- Comments on guarantees for solving the constrained problem [**Reviewers KSs2, Haeo, cVd6**]
- Requested clarifications from reviewer [**Reviewer Haeo**]

For the convenience of the reviewers, we highlight changes to the main body of the manuscript in the submitted pdf.

---

### Meta-Review · Area_Chair_noM8 · 2023-12-06

**Metareview:**

This paper presents a method for mitigating the impact on model accuracy during the pruning process. It introduces a constrained optimization problem solved via alternative gradient descent, focusing on decreasing the performance gap between average and worst-case subgroup performances, while maintaining overall accuracy.

Strengths
- Reviewers agree the method is novel and flexible.
- Experiments are sound and the implementation of the method is well described.
- Method outperforms existing alternatives on benchmarks like UTKFace and CIFAR100, without adding significant computational overhead.

Weaknesses
- The method's effectiveness with other pruning techniques is unclear. More diverse datasets, like ImageNet, could provide a broader validation.
- There are questions about the sensitivity of the algorithm to initialization and dual parameter updates. The feasibility and effectiveness of the constrained problem, especially in complex settings, need more clarification.
- The use of specific terminology (e.g., GMP vs IMP) could be clarified.

Overall, I recommend accept. The paper makes a solid contribution to pruning that may be of interest to readers.

**Justification For Why Not Higher Score:**

See weaknesses above.

**Justification For Why Not Lower Score:**

See strengths above.

---

### Decision · Program_Chairs · 2024-01-16

Accept (poster)